# Executive resources shape the impact of language predictability across the adult lifespan

**Merle Marie Schuckart[1,2]*[†], Sandra Martin[3][†], Sarah Tune[1,2], Lea-Maria Schmitt[4], Gesa Hartwigsen[3,5][‡], Jonas Obleser[1,2][‡]**

[1]Department of Psychology, University of Lübeck, Lübeck, Germany; [2]Center of Brain, Behavior and Metabolism, University of Lübeck, Lübeck, Germany; [3]Research Group Cognition and Plasticity, Max Planck Institute for Human Cognitive and Brain Sciences, Leipzig, Germany; [4]Donders Institute for Brain, Cognition and Behaviour, Radboud University, Nijmegen, Netherlands; [5]Wilhelm Wundt Institute for Psychology, Leipzig University, Leipzig, Germany

**\*For correspondence:**
merle.schuckart@uni-luebeck.de

[†]These authors contributed equally to this work
[‡]These authors also contributed equally to this work

## eLife Assessment

This study presents a **valuable** finding on whether executive resources mediate the impact of language predictability in reading in the context of aging. The evidence is **solid** in the investigation of prediction in reading, with one caveat that the text materials used could be biased against the aging population. The work will be of interest to cognitive neuroscientists working on reading, language comprehension, and executive control.

**Abstract** Humans routinely anticipate upcoming language, but whether such predictions come at a cognitive cost remains debated. In this study, we demonstrate the resource-dependent nature of predictive mechanisms in language comprehension across the lifespan: Experimentally limiting executive resources through a concurrent task reduces the effect of language predictability on reading time. Participants (*N* = 175, replication *N* = 96) read short articles presented word-by-word while completing a secondary font colour *n*-back task, thus varying cognitive demand. Language predictability was indexed by word surprisal as derived from a pre-trained large language model (GPT-2). Across two independent samples, our findings reveal that language predictions are not cost-free: They draw on executive control resources, and this dependency becomes more pronounced with age (18–85 years). These results help resolve the debate over cognitive demands in language comprehension and highlight prediction as a dynamic, resource-dependent process across the lifespan.

## Introduction

We constantly rely on our ability to swiftly yet accurately process linguistic input. When reading a book, watching television, or navigating a car through busy traffic while following instructions, language prediction is considered a catalyst that enhances the efficiency of linguistic processing (*Clark, 2013*; *Pickering and Garrod, 2007*; *Onnis et al., 2022*; *Rao and Ballard, 1999*; *Ryskin and Nieuwland, 2023*). However, due to the inherent flexibility and richness of natural language, upcoming words can rarely be predicted from context with complete certainty (*Rubenstein and Aborn, 1958*). Instead, linguistic features are thought to be pre-activated broadly rather than following an all-or-nothing principle, as there is evidence for predictive processing even for moderately or low-restraint contexts

**eLife digest** Understanding language requires more than simply recognizing individual words. Instead, readers and listeners continuously generate predictions about what might come next. For instance, after reading "She walked her…", most individuals would probably anticipate the word "dog."

Such predictive processes facilitate comprehension, making it faster and more efficient. Yet, a long-standing debate concerns whether these predictions emerge effortlessly and automatically, or whether they depend on domain-general executive resources, including working memory, attention, inhibitory control and goal-directed behaviour. Importantly, these resources are finite: they can be temporarily depleted when performing additional non-linguistic tasks and generally decline with age.

Schuckart et al. investigated whether language prediction during reading relies on executive resources and how language prediction might change with age as executive functioning declines. Employing a dual task reading paradigm, they examined how imposing a cognitively demanding secondary task influenced the effect of word predictability on reading speed in adults aged 18 to 85. This approach addressed a fundamental question in psycholinguistics – whether prediction is automatic or resource-dependent – and shed light on why language comprehension remains robust despite age-related cognitive decline.

The study involved 175 participants, with an independent replication sample of 96, who read texts word by word while sometimes performing a concurrent working-memory task. Word predictability was quantified using 'surprisal', a lexical score derived from the large language model GPT-2, which provides information about how unpredictable a word is.

The results revealed that less predictable words were read more slowly, confirming that readers actively generated linguistic predictions. The effect of predictability diminished under increased cognitive load, demonstrating that language prediction draws on executive resources. Older adults exhibited stronger predictability effects overall, but these were also more susceptible to reduction when executive resources were taxed. Together, the results show that language prediction is not cost-free and changes systematically across the adult lifespan.

These findings advance our understanding of how predictive language processing relies on executive resources and may have implications for interventions in conditions where these resources are compromised, such as following a stroke.

---

(*Boston et al., 2008*; *Roland et al., 2012*; *Schmitt et al., 2021*; *Smith and Levy, 2013*). This generation of graded predictions is sometimes described as being passive and cost-free (*Luke and Christianson, 2016*). However, it is still under debate whether maintaining such an elaborate process really incurs no cognitive cost.

Graded language predictions necessitate the active generation of hypotheses on upcoming words as well as the integration of prediction errors to inform future predictions (*Clark, 2013*; *Ryskin and Nieuwland, 2023*). Supporting this, recent evidence suggests that language predictions may indeed impose processing demands. *Shain et al., 2024* found that reading time increases with decreasing word predictability, with even small drops in predictability of highly expected words leading to significant processing costs. These findings suggest that language predictions are not entirely automatic or effortless. This aligns with numerous neuroimaging studies arguing for a strong interaction between language-specific and domain-general executive brain regions (*Geranmayeh et al., 2017*; *Martin et al., 2022*; *Sliwinska et al., 2017*; *Wingfield and Grossman, 2006*), particularly in situations that are cognitively demanding (*Erb et al., 2013*; *Vaden et al., 2013*; *Vaden et al., 2015*; *Vaden et al., 2016*; *Peelle, 2018*).

In this context, domain-general executive resources refer to higher-level cognitive control processes, such as working memory, inhibitory control, and cognitive flexibility, that are crucial for managing and coordinating behaviour across a wide range of tasks and modalities (*Alvarez and Emory, 2006*; *Duncan, 2010*; *Friedman and Miyake, 2017*). These processes are supported by the multiple-demand system, a fronto-parietal network that is recruited in various cognitively challenging situations (*Duncan, 2010*). However, while numerous studies support the claim that language prediction relies on such domain-general executive resources, a body of research suggests the opposite

(e.g., *Ryskin et al., 2020*; *Shain et al., 2020*; *Wehbe et al., 2021*; *Diachek et al., 2020*). This raises the unresolved question to what extent such resources are taxed by predictive processes (*Ryskin and Nieuwland, 2023*; *MacGregor et al., 2022*; *Shain et al., 2022*; *Xie et al., 2023*).

An interesting test case for the dynamic interplay between language-specific and domain-general systems is cognitive ageing, as advancing age has been shown to be associated with sensory and executive decline (*Idowu and Szameitat, 2023*; *Salthouse et al., 2003*; *Verhaeghen et al., 2003*). In the current study, we thus ask: How does language prediction change when executive resources are limited – both intrinsically due to advanced age, and extrinsically through increased task demands?

The age-related change in cognitive resources is reflected by longer linguistic processing times, especially in situations with high cognitive effort, such as dual-task processing (*Liu et al., 2016*; *Smiler et al., 2003*). However, previous research presents conflicting results on how cognitive ageing affects the use of linguistic predictions during language comprehension. Behavioural studies suggest a decline in using context to make semantic predictions with age (*Häuser et al., 2019*) while EEG studies present a more nuanced picture. Some studies indicate heightened neural sensitivity to unexpected information in older adults (*Cheimariou et al., 2019*), while others report no significant age-related differences in neural response (*Dave et al., 2018*; *Payne and Federmeier, 2018*).

Furthermore, it is unclear how the use of language predictions across the adult lifespan might be affected by increasing cognitive demands. According to the *Compensation-Related Utilisation of Neural Circuits Hypothesis* (CRUNCH; *Reuter-Lorenz and Cappell, 2008*), older adults might earlier reach a point where their cognitive load capacity is fully exhausted than young adults, leading to a performance decline. If language prediction draws on executive resources, its effect on reading time might thus diminish with increasing cognitive load due to shared cognitive resources. However, this effect might re-emerge once capacity limits are reached, causing tasks to be processed sequentially, which would result in an increased effect of language prediction paralleled by slower performance.

Here, we explored the role of executive control in language prediction in two large cohorts across the adult lifespan. Using a novel dual-task paradigm that couples natural reading with an *n*-back task that taxes executive resources, we tested the following hypotheses: First, both increased cognitive load and reduced word predictability (i.e., increased surprisal) should be reflected by longer reading time (*Figure 1a, b*), alongside a decrease in text comprehension and *n*-back task performance. Second, we expected that the formation of language predictions should be contingent upon the availability of executive resources. Most importantly, a gradual limitation of these resources due to increased task demands should result in diminished effects of language predictability on reading time (interaction between cognitive load and surprisal; *Figure 1c*).

Lastly, we explored how these previously described effects would be modulated by age (*Figure 1d*). Note that the literature allows for contradicting hypotheses: On the one hand, if language predictions are impaired under limited executive resources, older individuals should rely less on language predictions due to overall decreased executive resources (*Idowu and Szameitat, 2023*; *Salthouse et al., 2003*). This should be reflected by diminished predictability effects. On the other hand, given the presumed stability of language comprehension across the lifespan (*Shafto and Tyler, 2014*), older adults might instead rely more heavily on language predictions, thus fully compensating for any impairments in reading comprehension caused by sensory and executive decline (*Idowu and Szameitat, 2023*; *Salthouse et al., 2003*; *Verhaeghen et al., 2003*). In this case, we should see strong surprisal effects independent of age, or even stronger surprisal effects in older than younger adults.

The two large-sample reading-time studies presented here help resolve the ongoing discussion on the role of executive resources in language predictions with three key findings: First, across our large age range, we found a general increase in reading time with both low language predictability and high task demands. Second, higher task demands reduce the influence of language predictability on reading time, indicating that linguistic prediction relies on executive control resources. Third, predictability had a more pronounced effect on reading time in older adults compared to younger ones. These findings highlight the dynamic interaction between language predictability and executive resources.

## Results

We report data from 175 participants ($M$ = 44.9 ± 17.9, 18–85 years, 51% female) who were either tested online ($N_{online}$ = 80) or in the laboratory ($N_{laboratory}$ = 95) in one session. Moreover, we conducted

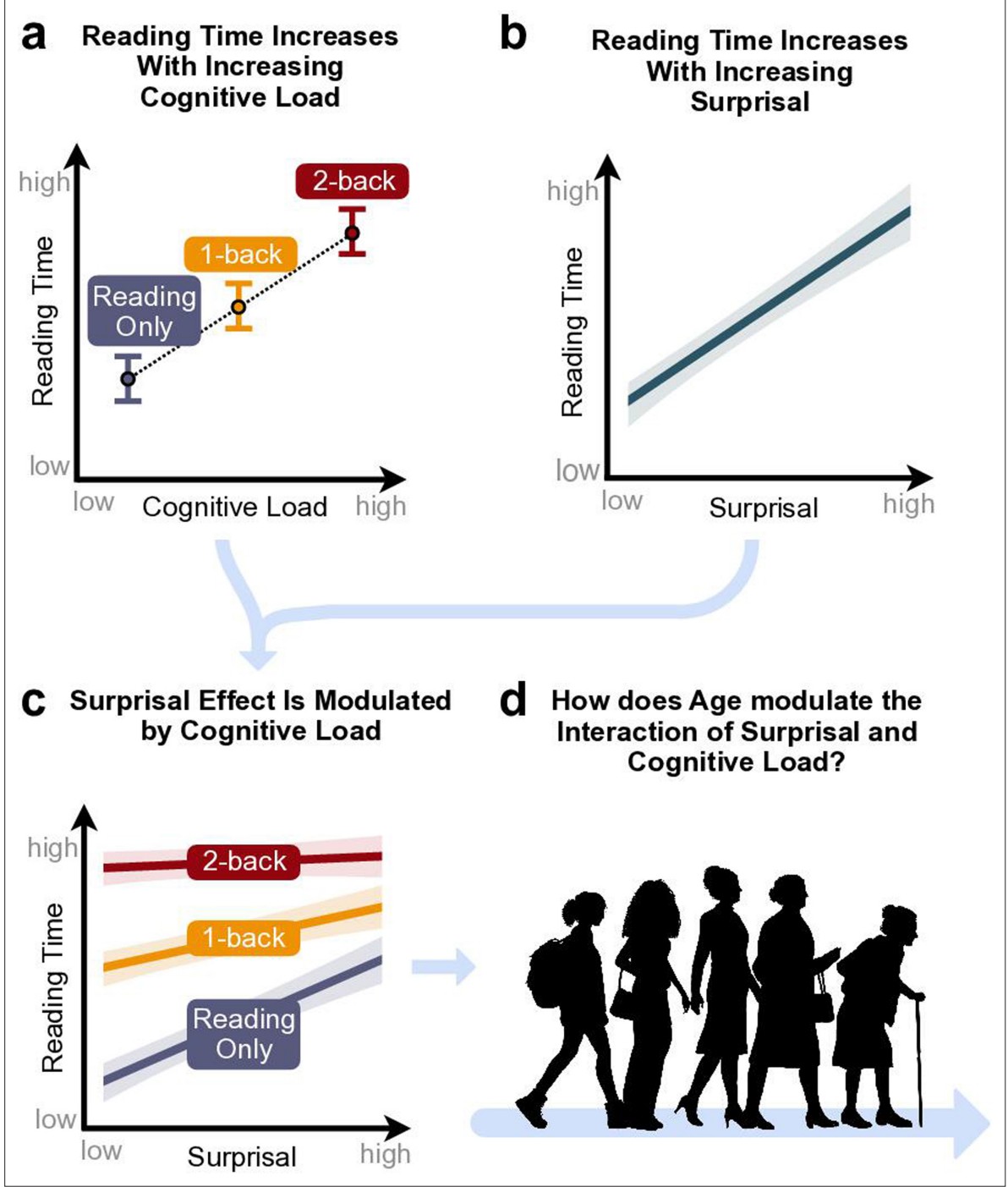

**Figure 1.** Visualisation of hypotheses. We expected main effects on reading time of (**a**) cognitive load and (**b**) surprisal, as well as (**c**) an interaction of surprisal and cognitive load. Additionally, (**d**) we explored how these effects are modulated by age.

an internal, pre-registered replication study involving another 96 participants ($M = 39.8 \pm 14.0$, 18–70 years, 51% female) tested online to replicate our main findings (see **Figure 2—figure supplement 1** for age distributions of all samples). During the experiment, participants engaged in a self-paced reading task. They read 300-word newspaper articles, presented word-by-word in various font colours (**Figure 2a**). The task was either performed in isolation (Reading Only) or paired with a competing *n*-back task on the words' font colour (1-back and 2-back Dual Task). Participants were

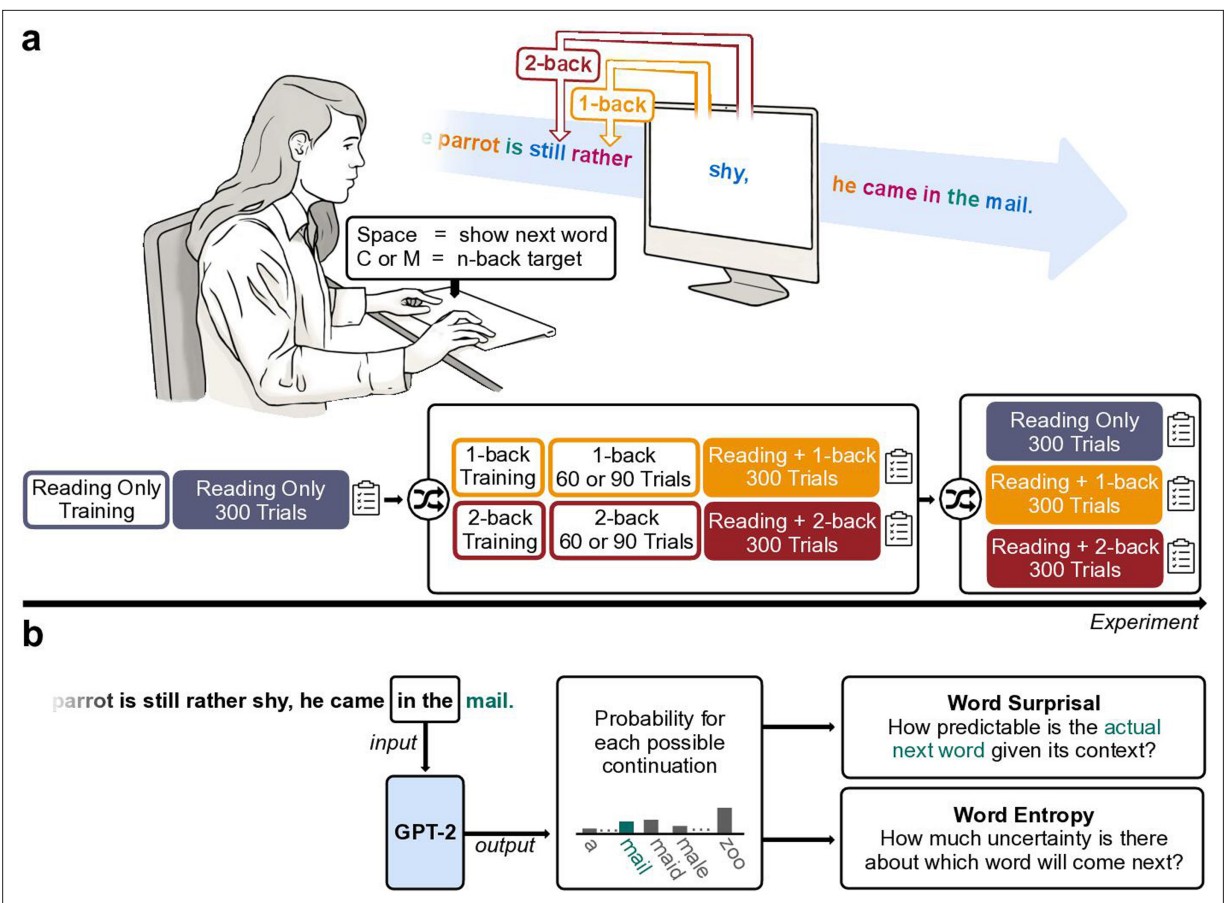

**Figure 2.** Experimental design and quantification of predictability as word surprisal using a large language model (GPT-2). (**a**) Participants were asked to perform a self-paced reading task (Reading Only) which was complemented in some blocks by a secondary *n*-back task on the font colour of the words (Reading + 1-back, and Reading + 2-back). The order of the blocks was pseudo-randomised, with Reading Only always being the first condition to be presented, followed by the two dual-task conditions, and another main block for each of the three conditions. Both dual-task paradigms (Reading + 1-back and Reading + 2-back) were first introduced in short single-task training sets. (**b**) We generated one surprisal score for each word in the reading material by using context chunks of two words as prompts for next-word predictions in GPT-2. The resulting probability for the actual next word in the text (here: 'mail', marked in teal) was then transformed into a surprisal score, which reflected how predictable the respective word was given the context. Additionally, based on the distribution of probabilities for all possible continuations, we computed an entropy score, which reflects the uncertainty in predicting the next word. Please note that the example sentence used here has been translated to English for better comprehensibility, while the original text materials were in German.

The online version of this article includes the following figure supplement(s) for figure 2:

**Figure supplement 1.** Comparison of age distribution between samples.

instructed to read the texts carefully, as content-related multiple-choice questions were asked after each text (see Methods for details).

## Increased cognitive load and older age reduce task performance

Results from a generalised mixed model for text comprehension accuracy showed that participants read the texts carefully and answered most comprehension questions correctly, with accuracies of 93% ± 14% (mean ± SD) in the Reading Only condition, and 80% ± 25% in the 1-back and 73% ± 28% in the 2-back Dual Task conditions (*Figure 3—figure supplement 1b*). Increases in cognitive load ($OR_{\text{1-back vs. BL}} = 0.253$, $z_{\text{1-back vs. BL}} = -8.928$, $p_{\text{1-back vs. BL}} < 0.001$, $OR_{\text{2-back vs. BL}} = 0.156$, $z_{\text{2-back vs. BL}} = -12.403$, $p_{\text{2-back vs. BL}} < 0.001$) and in age ($OR = 0.986$, $z = -2.676$, $p = 0.013$) were associated with poorer performance (see Methods for model details; *Figure 3a*, *Appendix 1—table 1*).

Similarly, $d'$ values demonstrated good performance in the *n*-back task, with mean $d'$ of 3.77 ± 0.8 in the 1-back and 2.12 ± 0.87 in the 2-back Dual Task condition (*Figure 3—figure supplement 1a*). The overall high $d'$ values observed here can be attributed to the low target ratio in the experiment,

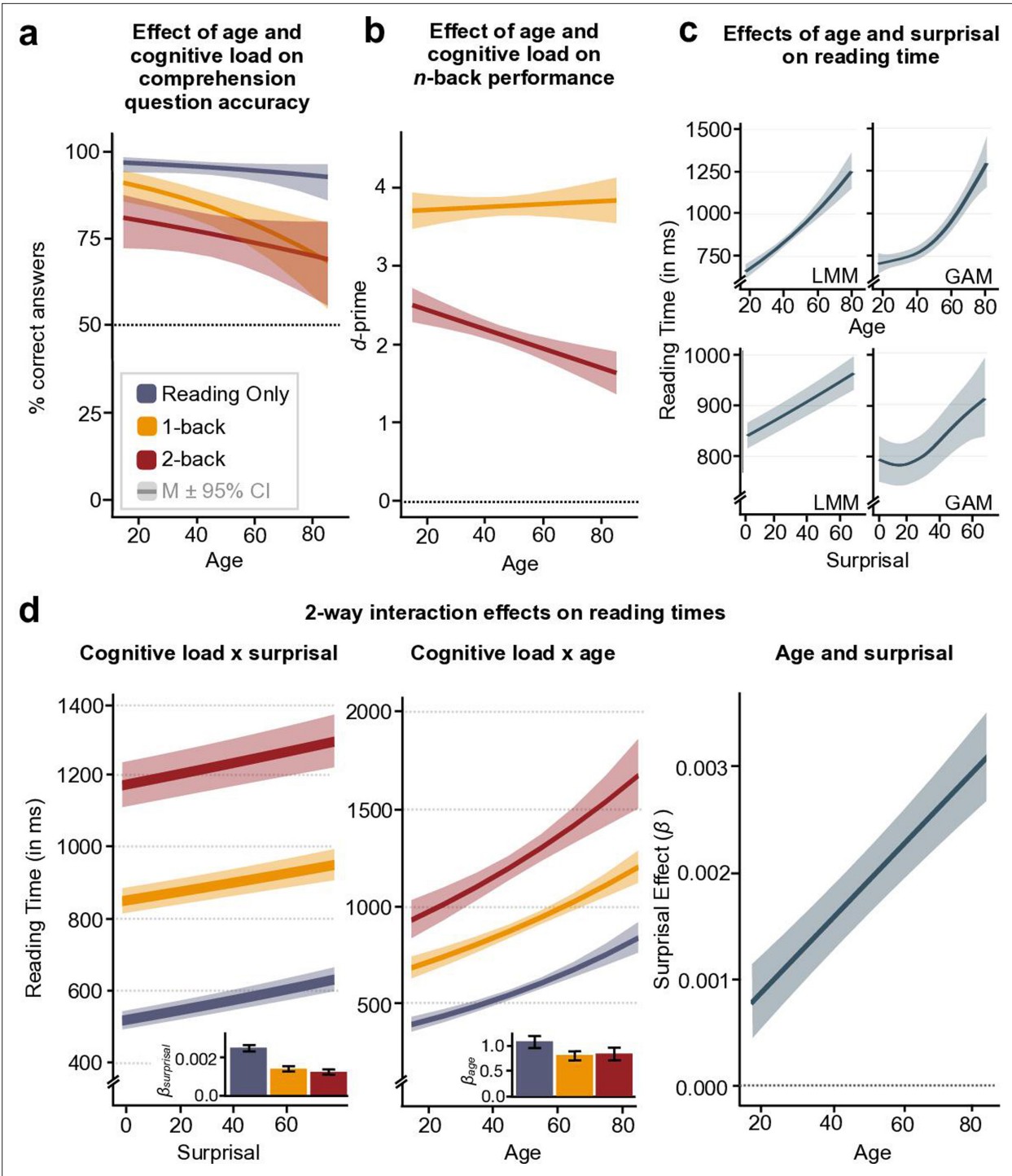

**Figure 3.** Estimated marginal effects of predictors age, cognitive load, and surprisal on task performance and reading time. Main effects of cognitive load and age on accuracy in the comprehension question task (**a**) and on *n*-back task performance (*d*-primes; **b**). Please note that we do not show *d*-primes for the Reading Only task as there was no *n*-back task in this condition. Reading time increased with increasing age and word surprisal (**c**, left: results from linear mixed model, LMM, right: results from generalised additive model, GAM – for an explanation see section *Modelling potential non-linear contributions*). In (panel **d**), we show the two-way interaction of cognitive load and surprisal (left) and cognitive load and age (middle). In both cases, effects were strongest in the Reading Only condition (see bar plot insets). Additionally, we show how age modulates the effect of surprisal on reading time (**c**, right). For raw and predicted individual trajectories, please see *Figure 3—figure supplements 1 and 2* in the Supplementary Material. Estimated marginal effects were adjusted for 'Reading Only' as the reference level. N = 175.

The online version of this article includes the following figure supplement(s) for figure 3:

*Figure 3 continued*

**Figure supplement 1.** Task performance and reading time by age and cognitive load condition.

**Figure supplement 2.** Individual predicted reading time.

resulting in high correct rejection and low false alarm rates. A linear mixed-effects model revealed *n*-back performance declined with cognitive load ($\beta$ = –1.636, *t*(173.13) = –26.120, *p* < 0.001), with more pronounced effects with advancing age ($\beta$ = –0.014, *t*(169.77) = –3.931, *p* > 0.001; ***Figure 3b***, ***Appendix 1—table 1***).

## Reading time increases with age, surprisal, and cognitive load

To operationalise language predictability, we calculated word surprisal scores using a 12-layered generative pre-trained transformer model (GPT-2; ***Radford et al., 2019***; ***Figure 2b***). Word surprisal quantifies the predictability of the current word given its preceding context (***Smith and Levy, 2013***; ***Hale, 2001***; ***Levy, 2008***). We chose a context length of two words, as constraining the context has been shown to increase GPT-2's psychometric predictive power, making its next-word predictions more human-like (***Kuribayashi et al., 2022***). In addition to each word's surprisal, we also computed word entropy, which reflects the uncertainty in predicting the next word. Thus, while word surprisal indicates the predictability of each word, word entropy reflects the uncertainty underlying its prediction.

We used linear mixed-effects regression (LMM) to assess the effect of word surprisal on reading time and its interaction with cognitive load and age (see *Methods* for model details). Our model as a whole was able to explain 65% of variance (conditional $R^2$) in single-word reading time from surprisal, *n*-back load, *n*-back performance, and other linguistic and demographic variables, and up to 81% when additionally considering the variability across subjects (marginal $R^2$; see ***Appendix 1—table 2***).

As hypothesised, we observed significantly longer reading time with advancing age, independent of cognitive load condition ($\beta$ = 0.009, *t*(178.46) = 9.199, *p* < 0.001). For illustration, each additional year of age increased reading time by roughly 1%, respectively (see ***Figure 3c***, top; ***Table 1***; ***Figure 3—figure supplement 2a***).

To account for the potential influence of verbal intelligence and education on the observed age effects, we compared three additional LMMs: a baseline model without these predictors, and two additional models, one including a verbal intelligence predictor and the other including an education predictor (please see Methods section *Control analysis: Assessing potential effects of verbal intelligence and education*).

Adding verbal intelligence as a predictor to our model did not significantly improve the model fit ($\chi^2(1)$ = 0.769, *p* = 0.381). This implies that verbal intelligence cannot account meaningfully for the differences in reading time found between younger and older participants.

When including years of education as a predictor, we observed a significant effect on reading time ($\beta$ = 0.009, *t* = 9.199, *p* < 0.001), which led to a modest yet significant improvement in model fit ($\chi^2(1)$ = 4.209, *p* = 0.0402, $AIC_{model\_baseline}$ = 37,580, $AIC_{model\_education}$ = 37,578). However, the three-way interaction of age, surprisal and cognitive load we found in our original model remained significant ($\beta_{1\text{-back}}$ = –0.00007, $t_{1\text{-back}}$(88.61) = –5.229, $p_{1\text{-back}}$ < 0.001, $\beta_{2\text{-back}}$ = –0.00004, $t_{2\text{-back}}$(88.90) = –2.784, $p_{2\text{-back}}$ = 0.007), suggesting that while education has a significant effect on reading time, it cannot account for the age-related effects observed in our models.

Analogous to the effect of age, increased cognitive load was associated with significantly longer reading time ($\beta_{1\text{-back}}$ = 0.474, $t_{1\text{-back}}$(176.18) = 34.046, $p_{1\text{-back}}$ < 0.001, $\beta_{2\text{-back}}$ = 0.792, $t_{2\text{-back}}$(173.76) = 30.339, $p_{2\text{-back}}$ < 0.001; ***Table 1*** and ***Figure 3—figure supplements 1 and 2***), indicating participants read more slowly when faced with a more challenging task. Even after excluding the Reading Only condition from the model and comparing only the two equally attention-demanding dual-task conditions 1-back and 2-back (control analysis), hereby controlling for attentional switching costs, this effect still held true ($\beta$ = 0.339, *t*(178.98) = 16.221, p <0 .001; ***Appendix 1—table 4***).

Moreover, in line with our hypothesis, we observed a consistent increase in reading time with higher surprisal ($\beta$ = 0.002, *t*(2361.37) = 11.321, *p* < 0.001; ***Table 1*** and ***Figure 3c***, bottom). Highly predictable words were read more quickly. Specifically, for a change in surprisal by one standard deviation, reading time increased by about 2.1%. For words and individuals matched in all other regards, this translates to a mean reading-time difference of approximately 118 ms between words with the minimum and maximum surprisal values in our dataset (range of surprisal values: 3.56–72.19).

**Table 1.** Main results for model for reading time ($N$ = 175).

| | Predictors | Estimate | Std. error | CI | t | df | p | |
|---|---|---|---|---|---|---|---|---|
| **Main effects** | Surprisal | 0.001707 | 0.000151 | 0.001411 to 0.002002 | 11.320677 | 2361.37 | $1.368 \times 10^{-28}$ | * |
| | Age | 0.009113 | 0.000991 | 0.007158 to 0.011068 | 9.199100 | 178.46 | $1.751 \times 10^{-16}$ | * |
| | Cognitive load [1-back vs. Reading Only] | 0.473800 | 0.013916 | 0.446336 to 0.501264 | 34.046321 | 176.18 | $8.399 \times 10^{-79}$ | * |
| | Cognitive load [2-back vs. Reading Only] | 0.791540 | 0.026090 | 0.740046 to 0.843034 | 30.338989 | 173.76 | $7.320 \times 10^{-71}$ | * |
| | Surprisal × age | 0.000035 | 0.000004 | 0.000027 to 0.000042 | 9.287151 | 287,771.27 | $3.481 \times 10^{-20}$ | * |
| | Surprisal × cognitive load [1-back vs. Reading Only] | −0.001093 | 0.000161 | −0.001409 to −0.000776 | −6.771521 | 287,959.11 | $2.043 \times 10^{-11}$ | * |
| | Surprisal × cognitive load [2-back vs. Reading Only] | −0.001255 | 0.000163 | −0.001575 to −0.000935 | −7.681261 | 288,294.96 | $2.709 \times 10^{-14}$ | * |
| | Age × cognitive load [1-back vs. Reading Only] | −0.002798 | 0.000776 | −0.004330 to −0.001267 | −3.606479 | 171.99 | $5.135 \times 10^{-4}$ | * |
| **Two-way interactions** | Age × cognitive load [2-back vs. Reading Only] | −0.002458 | 0.001454 | −0.005329 to 0.000412 | −1.690400 | 170.79 | $9.681 \times 10^{-2}$ | |
| | Surprisal × age × cognitive load [1-back vs. Reading Only] | −0.000111 | 0.000009 | −0.000129 to −0.000094 | −12.266076 | 287,807.34 | $3.748 \times 10^{-34}$ | * |
| **Three-way interactions** | Surprisal × age × cognitive load [2-back vs. Reading Only] | −0.000078 | 0.000009 | −0.000096 to −0.000060 | −8.483676 | 287,771.65 | $4.384 \times 10^{-17}$ | * |
| **Model fit** | Intra-class correlation (ICC) | 0.46 | | | | | | |
| | Marginal $R^2$/conditional $R^2$ | 0.643/0.807 | | | | | | |

All continuous predictors were centred. Degrees of freedom for p-values, standard errors and confidence intervals (CIs) were computed using Satterthwaite's approximation. All p-values reported here are FDR-corrected and were computed using ANOVAs with type III sum of squares. Results that are significant on an alpha-level of 0.05 are marked with a star.

## Cognitive load reduces the impact of surprisal

In line with our hypotheses, cognitive load significantly modulated the effect of surprisal on reading time ($\beta_{1\text{-back}}$ = –0.001, $t_{1\text{-back}}$(287,959.11) = –6.772, $p_{1\text{-back}}$ < 0.001, $\beta_{2\text{-back}}$ = –0.001, $t_{2\text{-back}}$(288,294.96) = –7.681, $p_{2\text{-back}}$ < 0.001). While we observed a clear increase in reading time in the Reading Only condition when surprisal was high, this effect was mitigated in both the 1-back and 2-back Dual-Task condition, where cognitive load was increased (*Table 1* and *Figure 3d*, left).

## Age modulates the effect of cognitive load

While increased task demands were associated with prolonged reading time across participants, we found this effect became less pronounced with advancing age ($\beta_{1\text{-back}}$ = –0.003, $t_{1\text{-back}}$(171.99) = –3.606, $p_{1\text{-back}}$ < 0.001; $\beta_{2\text{-back}}$ = –0.002, $t_{2\text{-back}}$(170.79) = –1.690, $p_{2\text{-back}}$ = 0.097; *Table 1* and *Figure 3d*, middle). To illustrate, when comparing the reading time in the 2-back Dual Task and the Reading Only condition, we found an increase of 130.48% in an average young (i.e., 27 years old) and an increase of 111.30% in an average older (i.e., 63 years old) participant.

## Age modulates the effect of surprisal

Older age was associated with stronger surprisal effects ($\beta$ = 0.00004, $t$(287,771.27) = 9.287, $p$ < 0.001; *Table 1* and *Figure 3d*, right), indicating that highly unpredictable (i.e., surprising) words were associated with a significantly longer reading time, especially in older adults. This suggests that, as individuals age, the effect of word predictability on their reading time becomes increasingly pronounced instead of remaining constant or diminishing.

## Age modulates the interaction of surprisal and cognitive load

Finally, we tested whether the observed interaction between cognitive load and surprisal changes with advancing age: We found that age indeed modulated the joint influence of cognitive load and surprisal on reading time, as reflected by a significant three-way interaction of surprisal, cognitive load and age ($\beta_{1\text{-back–Reading Only}}$ = –0.00011, $t_{1\text{-back–Reading Only}}$(287,807.34) = –12.2661, $p_{1\text{-back–Reading Only}}$ < 0.001, $\beta_{2\text{-back–Reading Only}}$ = –0.00008, $t_{2\text{-back–Reading Only}}$(287,771.65) = –8.484, $p_{2\text{-back–Reading Only}}$ < 0.001; *Table 1*).

Even after excluding the Reading Only condition from the model and contrasting only the dual-task conditions 1-back and 2-back (control analysis), results still showed a significant three-way interaction of age, cognitive load, and surprisal ($\beta_{2\text{-back–1-back}}$ = 0.00003, $t$(188,203.53) = 3.373, $p$ = 0.001; *Appendix 1—table 4*). Please note the change in reference level caused by the exclusion of the Reading Only condition, which causes a reversal in the direction of the three-way interaction between age, cognitive load, and surprisal.

To get a more nuanced understanding of age-related differences in the effect of cognitive load and surprisal on reading time, we conducted a simple slopes analysis for our original model (*Figure 4*, *Figure 4—figure supplement 1*): Under low cognitive load (condition Reading Only), surprisal significantly influenced reading time in all but the youngest participants. Put simply, when participants read a text without having to perform an additional n-back task, predictable (i.e., low surprisal) words yielded significantly shorter reading time than unpredictable (i.e., high surprisal) words (*Figure 4a*, left panel, first plot). This effect was most pronounced in the oldest participants ($\beta_{85-18 \text{ years}}$ = 0.006, $p$ < 0.001).

As task demands increased, we observed a reversal of this pattern, with younger participants exhibiting stronger surprisal effects than older participants (condition 1-back; *Figure 4a*, left panel, second plot). Notably, under increased cognitive load, even the youngest participants started showing significant surprisal effects ($\beta_{85-18 \text{ years}}$ = –0.0009, $p$ = 0.04).

Finally, in the most demanding condition (2-back), the pattern shifted again, with older adults again showing stronger surprisal effects than their younger counterparts ($\beta_{85-18 \text{ years}}$ = 0.001, $p$ = 0.003; *Figure 4a*, left panel, third plot).

Comparing surprisal effects between cognitive load conditions revealed that older adults showed the most pronounced reduction in surprisal effects as cognitive load increased (*Figure 4b*), which suggests they were more vulnerable to increased task demands than younger participants (18 year-old: $\beta_{1\text{-back–Reading Only}}$ = –0.003, $p$ < 0.001, $\beta_{2\text{-back–Reading Only}}$ = –0.003, $p$ < 0.001, $\beta_{2\text{-back–1-back}}$ = 0.0004, $p$ = 0.06; 85 year-old: $\beta_{1\text{-back–Reading Only}}$ = –0.011, $p$ < 0.001, $\beta_{2\text{-back–Reading Only}}$ = –0.008, $p$ < 0.001, $\beta_{2\text{-back–1-back}}$ = 0.003, $p$ < 0.001).

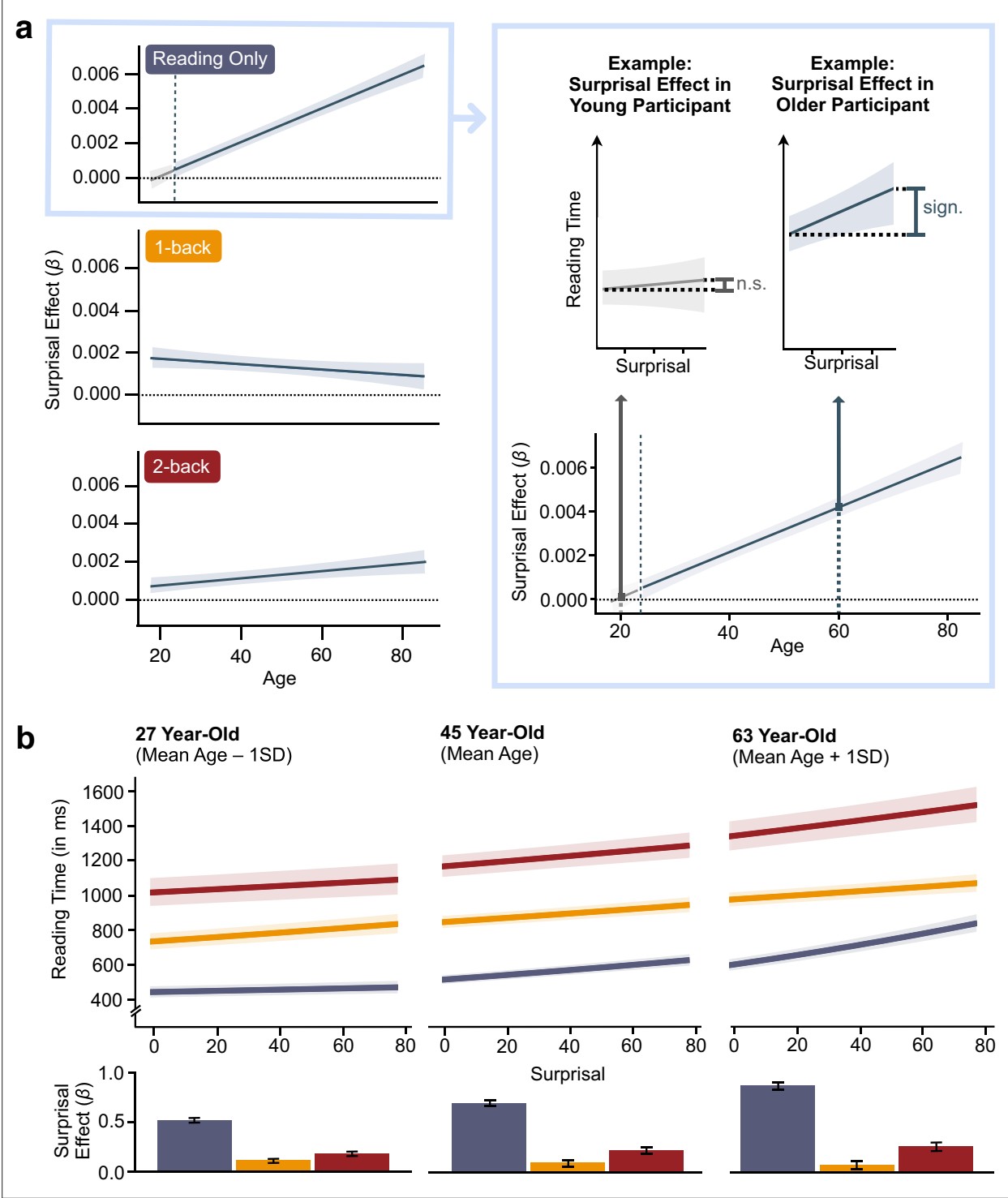

**Figure 4.** Results of the simple slopes analysis and exemplary marginal effects plots for three different ages. In the Johnson–Neyman plot (*Johnson and Neyman, 1936*) on the left side of panel (**a**), we show the effect of surprisal on reading time across the whole age range separated by cognitive load condition: *Reading Only* (top; blue), *1-back Dual Task* (middle; yellow), and *2-back Dual Task* (bottom; red). The stronger the surprisal effect for a certain age, the higher the value on the *y*-axis. Grey areas indicate age ranges for which we did not find an effect of surprisal on reading time in the respective condition, whereas blue areas indicate a significant surprisal effect (see inset on the right for a visualisation of a non-significant effect in a younger participant and a significant effect in an older participant). In panel (**b**), we show the predicted surprisal effect in each cognitive load for an average young (average age −1 SD), middle-aged (average age) and older participant (average age +1 SD). The bar plots illustrate the predicted effects of surprisal on reading time (Estimates ± 95% CI) across the three cognitive load conditions for those three average participants. N = 175.

*Figure 4 continued on next page*

*Figure 4 continued*

The online version of this article includes the following figure supplement(s) for figure 4:

**Figure supplement 1.** Comparison of factor smooths for different levels of cognitive load from the three-way interaction of age, surprisal, and cognitive load.

**Figure supplement 2.** Comparison of the results of the LMM and GAM control analyses (Estimates ± 95% CI).

**Figure supplement 3.** Estimates ± 95% CI for the three-way interaction of age, entropy, and cognitive load in the full sample (*N* = 175).

## Modelling potential non-linear contributions

To account for the possibility that age, surprisal, and their interaction with cognitive load might demonstrate non-linear effects on reading time, we fitted a generalised additive mixed-effects model (GAM) to our data, using the same model structure as for the linear regression model (see Methods for model details). Results from the GAM revealed overall similar effects relative to the LMM. The effective degrees of freedom (EDF) of all continuous predictors were estimated above 1, confirming their non-linearity. Similar to the LMM, both our predictors of interest, surprisal and age, demonstrated significant effects on reading time (*Figure 3c*, right), with surprisal showing a more non-linear trajectory than age (EDF for surprisal: 4.107, EDF for age: 3.028, both *p* < 0.001). The smoothing splines for the three-way interaction of age, surprisal, and cognitive load showed significant effects for all levels of cognitive load (*Figure 4—figure supplements 1 and 2*). Similar to the LMM, we found the strongest effect for the Reading Only condition (EDF for Reading Only: 10.248, p < 0.001, EDF for 1-back: 2.017, *p* = 0.036, EDF for 2-back: 4.89, *p* = 0.024). The full model results are reported in *Appendix 1—table 5* and *Figure 4—figure supplement 1*.

## Disentangling the effect of cognitive load on pre- and post-stimulus predictive processing

The present study focuses primarily on surprisal, an information-theoretic measure of how unexpected an encountered word is given its preceding context. As such, it reflects the post-stimulus integration of predictions. In an additional control analysis, we considered entropy, which reflects the uncertainty (and thus the inverse precision) of pre-stimulus predictions, or – put simply – the expected surprisal of an upcoming word (*Pimentel et al., 2022*). To examine whether limitations in executive resources constrain prediction generation before word onset or prediction integration after word onset, we modelled the effects of entropy on reading times, using the same approach as for modelling the effect of surprisal. Analogous to the effects of surprisal, we expected reading time to increase with higher entropy and anticipated that increased cognitive load would attenuate the effect of entropy on reading time.

Contrary to our hypothesis, results indicated that in the Reading Only condition, with minimal cognitive load, participants benefitted from higher entropy, as reflected by shorter reading times. As the reading materials were designed to be as easy to understand as possible, entropy was overall very low, ranging from 0.708 to 6.763 (see *Bentz and Alikaniotis, 2016* for an overview of entropy in natural texts in different languages). The observed facilitated processing for segments with slightly increased entropy is thus consistent with previous findings that entropy can have facilitating effects on language comprehension due to pre-activation of semantic features (*Karimi et al., 2024*).

Interestingly, as cognitive load increased, this effect reversed: participants showed longer reading times when entropy was high ($\beta_{1\text{-back}}$ = −0.0066, $t_{1\text{-back}}$(287,500.04) = 5.214, $p_{1\text{-back}}$ < 0.001, $\beta_{2\text{-back}}$ = 0.0065, $t_{2\text{-back}}$(287,757.94) = 5.026, $p_{2\text{-back}}$ < 0.001), which is in line with our hypotheses. While we found this pattern – beneficial effect of entropy under minimal cognitive load and detrimental effect under increased cognitive load – in our younger participants, older participants showed a detrimental effect of increased entropy across conditions. Moreover, paralleling the results for surprisal, older adults were more sensitive to variations in entropy than their younger counterparts ($\beta_{1\text{-back}}$ = −0.00004, $t_{1\text{-back}}$(287,440.36) = −5.547, $p_{1\text{-back}}$ < 0.001, $\beta_{2\text{-back}}$ = −0.00002, $t_{2\text{-back}}$(287,488.32) = −2.578, $p_{2\text{-back}}$ < 0.001; see and *Figure 4—figure supplement 3* and *Appendix 1—table 6* for full results).

## Interaction of surprisal and cognitive load generalises to new sample

To probe the reliability of our findings, we conducted an exact online replication of the original experiment. The replication model included the main effects of age, cognitive load, and surprisal, and the

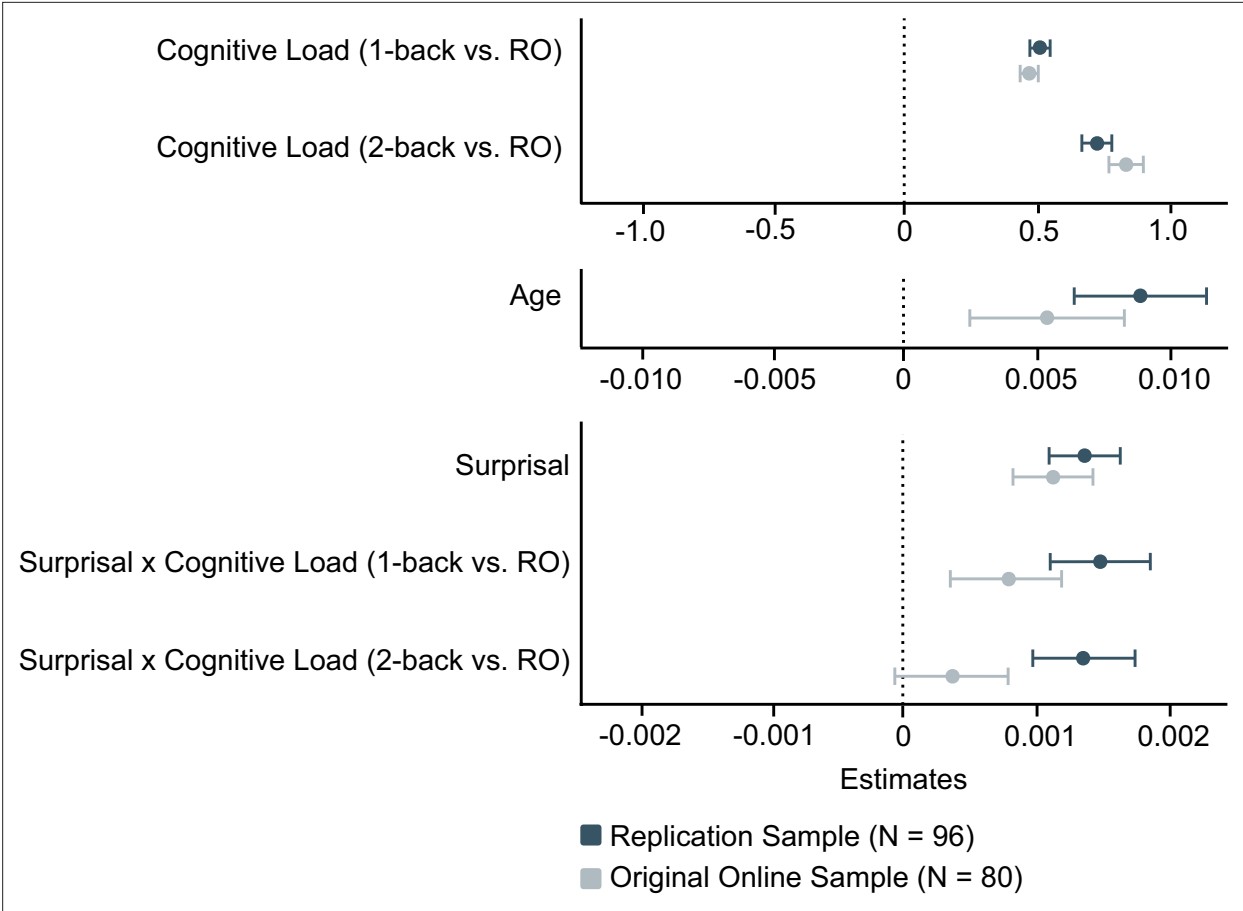

**Figure 5.** Results of the internal online replication in comparison with the results of the online sample of the original study. Estimates ±CI for the main effects of age, surprisal, and cognitive load as well as the two-way interaction of surprisal and cognitive load are visualised. RO: Reading only. Full results are provided in *Appendix 1—table 3*. For a comparison of age distributions in the original online and lab sample and the online replication sample, please see *Figure 2—figure supplement 1*. Please note that effects are grouped by their magnitude.

The online version of this article includes the following figure supplement(s) for figure 5:

**Figure supplement 1.** Estimates ± 95% CI for the three-way interaction of age, surprisal, and cognitive load in the replication sample (N = 96).

two-way interaction between cognitive load and surprisal; the three-way interaction with age was not modelled. This streamlined model ensured adequate statistical power and yielded stable, interpretable estimates given the available sample size. When comparing the results of the online participants from the original and replication samples, we found highly consistent effects (see *Figure 5*). Again, a significant interaction of cognitive load and surprisal emerged ($\beta_{\text{1-back}}$ = 0.001499, $t_{\text{1-back}}$(161,262.31) = 7.377, $p_{\text{1-back}}$ < 0.001; $\beta_{\text{2-back}}$ = 0.001365, $t_{\text{2-back}}$(161,923.06) = 6.721, $p_{\text{2-back}}$ < 0.001; see *Appendix 1— table 3* and *Figure 4—figure supplement 2*), despite the smaller sample size (N = 96 vs. 175).

## Modelling cumulative effects of surprisal on reading time

As noted above (see section *Reading time increases with age, surprisal, and cognitive load*), reading time increased as a function of word surprisal, with a mean difference of 118 ms between the most and least predictable word in our text material. This corresponds to approximately 22.91% of the average per-word reading time in the BL condition (M = 522.04 ± 275.927 ms), highlighting a substantial effect of surprisal – especially considering that all other predictors were held constant when estimating the effect of surprisal. The substantial effect of surprisal is particularly notable given that the texts were edited for ease of comprehension and contained relatively low surprisal values overall (M = 18.165 ± 7.523), indicating that the observed reading time differences between high- and low-surprisal words likely underestimate the potential effect size in more complex texts with higher surprisal variability.

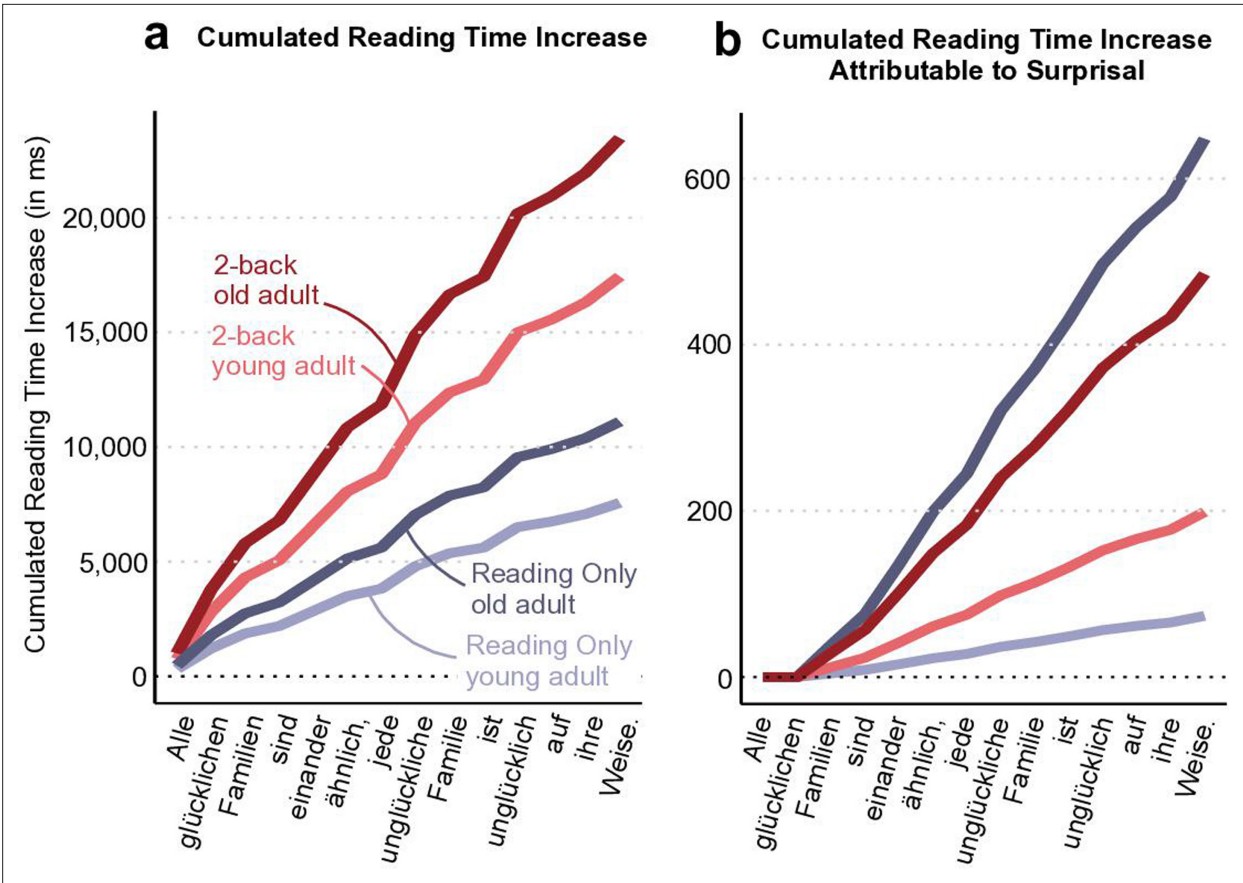

**Figure 6.** Estimates ± 95% CI for the cumulative effect of surprisal on reading time. To illustrate the cumulative effect of surprisal on reading time over the course of a text, we predicted reading times for an average younger (27 years, $M - 1$ SD) and average older (63 years, $M + 1$ SD) participant in the easy Reading Only condition (blue) and the most challenging condition 2-back (Dual Task; red) and computed the cumulative sum for a short example sentence. Panel **a** illustrates how reading time gradually increases in total over the course of the sentence, with all predictors being held constant at their average, except for the predictors age, cognitive load, and word length. In panel **b**, we again show cumulative reading times, this time isolating the effect of surprisal. Please note that surprisal values are zero for the first two words, as our GPT-2 model estimates surprisal based on the two preceding words, which are unavailable at the beginning of the sentence. The example sentence used in both panels is the German translation of the opening line of *Anna Karenina*, 'Happy families are all alike, every unhappy family is unhappy in its own way' (**Karenina, 1878**). N = 175.

Notably, in everyday life, we typically encounter sequences of words, ranging from short phrases to texts of hundreds or thousands of words. Consequently, small word-level effects can accumulate and yield substantial processing differences over time (cf. **Funder and Ozer, 2019**). Thus, to quantify this cumulative effect of surprisal, we predicted reading time for two average participants aged 27 ($M - 1$ SD) and 63 ($M + 1$ SD) years for a short example sentence. Predictions were conducted for the easiest cognitive load condition Reading Only, in which surprisal effects were most pronounced, and for the most challenging condition 2-back Dual Task, where surprisal effects were diminished. The example sentence comprised 14 words and had relatively low surprisal values ($M = 15.24 \pm 7.65$; even slightly lower than in our original text material), implying that the cumulative effects of surprisal shown here are rather conservative estimations.

In the condition with the largest surprisal effects (Reading Only), surprisal led to a total cumulative increase in reading time of 73.6 ms for younger and 648 ms for older participants over the course of the sentence. In the more challenging 2-back Dual Task condition, we observed a total increase of 199 ms in younger and 485 ms in older participants (see **Figure 6b** for cumulated effects of surprisal; see **Figure 6a** for predicted reading time incorporating all effects for a comparison).

## Discussion

Linguistic predictions are a powerful feature of language comprehension. But do they really come 'for free', or how much do they draw on executive resources? With the present study, we asked how language predictions change with increasing cognitive load, and how this interaction is modulated by age. To do so, we paired a self-paced reading task with a secondary *n*-back task on the font colour of the words.

First, as expected, and validating our overall approach, high cognitive load was associated with an increase in reading time as well as a decrease in task performance across age groups. This is consistent with previous studies using *n*-back tasks (*Lamichhane et al., 2020*; *Kwong See and Ryan, 1995*).

Next, as hypothesised, higher word surprisal slowed down reading, even when controlling for word length and frequency as well as prediction entropy. This effect of surprisal replicates findings from previous studies showing that highly unpredictable words are generally associated with a longer reading time (*Schmitt et al., 2021*; *Smith and Levy, 2013*; *Heilbron et al., 2022*; *Monsalve et al., 2012*; *Shain et al., 2024*; *Wilcox et al., 2023*).

Finally, we explored the relationship between surprisal and cognitive load across the (cross-sectional) adult lifespan. We hypothesised that increasing cognitive load should gradually impair the building of language predictions, which should surface in a diminished surprisal effect on reading time in conditions with high cognitive load. In line with our hypothesis, we found that the effect of word surprisal on reading time was modulated by cognitive load. Specifically, when cognitive load was high, the effect of surprisal on reading time was significantly diminished.

Interestingly, this interaction between surprisal and cognitive load was modulated by age. While age generally increased the reliance on language prediction, it also increased the susceptibility of this strategy to changes in available executive resources: Older adults showed the strongest relative reduction of the surprisal effect with increasing cognitive load. However, under high load, older adults still showed the strongest surprisal effect in absolute terms (*Figure 4b*). In a direct replication of the original experiment, we reproduced this finding, further confirming the reliability of our results.

### Disentangling effects of attention versus executive resources

When investigating the interaction of cognitive load and language predictability on reading time, we found that increased cognitive load diminished the effect of word predictability. We take this as evidence that executive resources are involved in the generation of language predictions.

Predictive processing has not only been suggested to be foundational to language comprehension (*Ryskin and Nieuwland, 2023*; *Schmitt et al., 2021*; *Heilbron et al., 2022*), but it is also thought to be a core mechanism of the human brain (*Rao and Ballard, 1999*; *Bubic et al., 2010*; *Friston, 2010*). Drawing parallels between domains can thus offer valuable insights into the mechanisms of language prediction: For instance, there is evidence from the visual domain showing that attending to the stimulus material is a prerequisite for predictive processing (*Larsson and Smith, 2012*; *Richter and de Lange, 2019*). This observation from the visual domain can potentially be extended to our findings, wherein language predictions were diminished if attention had to be divided between the reading material and a challenging non-linguistic secondary task.

However, attention alone cannot fully account for the differences in sensitivity to word predictability shown here. The predictability effect did not only diminish when a secondary task was introduced but also decreased with increasing cognitive load even when attentional switching costs were held constant between conditions (i.e., when comparing the 1-back to 2-back load conditions). In line with previous literature (*Huettig and Janse, 2016*; *Ito et al., 2018*; *Fricke and Zirnstein, 2022*), our results thus suggest that executive processes beyond attention, such as updating and maintenance of context information, shifting between tasks, and inhibition of irrelevant information (*Alvarez and Emory, 2006*; *Duncan, 2010*), are integral to language prediction.

### Language prediction as a compensatory mechanism in older age

Further examining the effect of language predictability across the adult lifespan revealed interesting age differences. Namely, older adults showed stronger language predictability effects than younger participants. This effect held true even when controlling for potential differences in verbal intelligence and education between participants. Our result aligns with previous work on age-related changes

in linguistic predictions, indicating heightened sensitivity to unexpected lexical input in older adults (*Cheimariou et al., 2019*).

This finding may reflect a commonly reported pattern of greater reliance on intact vocabulary and world knowledge with age in the face of declining executive functions (*Martin et al., 2022*; *Salthouse et al., 2003*). Here, we show that even under increased cognitive load, older adults still rely heavily on their refined system of linguistic predictions driven by their lifelong experience. This allows them to make more fine-tuned predictions but also renders them more vulnerable to unexpected information (i.e., high-surprisal words) than their younger counterparts.

Previous studies have found a larger vocabulary size associated with more rapid processing of language and improved language comprehension (*Borovsky et al., 2012*; *Huettig and Pickering, 2019*; *Mainz et al., 2017*; *Matthews, 2018*; *Stæhr, 2009*; *Stæhr, 2008*). In line with this, older adults exhibit more advanced language processing abilities compared to children or younger adults due to their accumulated years of exposure to language and their increased vocabulary (*Brysbaert et al., 2016*; *Ito and Sakai, 2021*). This accumulated skill is thought to serve as a compensatory mechanism for decline in working memory capacities or reduced executive functioning with age (*Bunzeck et al., 2024*; *Reuter-Lorenz et al., 2021*). Indeed, speech comprehension appears to remain largely intact in older adults (*Shafto and Tyler, 2014*; *Burke and Mackay, 1997*).

As a caveat, one should not disregard the possibility of age-related differences between younger and older adults in utilising formed predictions. While we assumed that formed predictions are utilised automatically, and that the observed differences in the effect of language predictability on reading time between individuals might thus be attributable to a difference in executive resources, one could also argue that individuals might simply weigh their formed predictions differently.

Akin to the longer exposure to language in older individuals, it is reasonable to assume that older individuals have also had more time to accumulate experience regarding the accuracy of their predictions and to refine their predictions through prediction errors. Consequently, older adults might rely more heavily on their predictions than younger adults (*Chan et al., 2021*; *Moran et al., 2014*). Additionally, due to age-related sensory decline (*Verhaeghen et al., 2003*), older adults might exhibit a stronger dependence on context-based predictions to process language, as incoming sensory information might be less informative (*Rogers, 2017*; *Wingfield et al., 2015*; *Wolpe et al., 2016*). A stronger reliance on language predictions could thus serve as a compensatory mechanism to facilitate language comprehension despite sensory decline in older adults.

## How can limited executive resources affect language prediction?

As shown in *Figure 4*, the relationship between word predictability and age depends on cognitive load. Under low load, predictability effects on reading time increased with age; older adults showed robust effects, while younger adults showed none. Under intermediate load (1-back), this pattern reversed, with younger adults showing stronger predictability effects than older adults.

This reversal begs for an explanation, and we deem it most likely to reflect differences in how executive resources are deployed. In low-load settings, young adults may process both predictable and unpredictable words efficiently, minimising observable surprisal effects. The absence of a predictability effect in this group should therefore not be taken as evidence against predictive processing. Rather, it may indicate that prediction is less necessary when processing is fast and flexible. Under intermediate load, executive resources are partially taxed, revealing underlying prediction processes in young adults. At higher loads, predictability effects diminish again, suggesting resource constraints impair predictive processing.

In older adults, however, predictability effects decline already at intermediate load, consistent with the CRUNCH model (*Reuter-Lorenz and Cappell, 2008*), which posits that cognitive capacity limits are reached earlier in ageing. At this point, resources are insufficient to maintain predictive processing while also performing the secondary task. Behaviourally, this may result in fluctuating performance or trial-wise switching between the two tasks. As load increases further, older adults continue to show reduced, though still measurable, predictability effects – indicating sustained but strained processing.

Taken together, the data suggest that both age groups experience a reduction in predictive processing when executive resources are limited, but the 'crunch point' is reached earlier in older adults.

## Limitations and future directions

One intriguing question that remains is how *n*-back performance and language surprisal interrelate. It is plausible that when texts are highly predictable (i.e., when surprisal is very low on average), the cognitive load associated with language processing is reduced. This reduction could free up domain-general executive resources, thereby enhancing *n*-back performance. Conversely, when surprisal is high, the increased demands of processing less predictable language may compromise working memory updating and lead to poorer *n*-back performance. As we assessed *n*-back performance at block-level and maintained equal predictability across texts, analysing trial-level effects of surprisal on *n*-back task performance was not possible. Future studies could address this limitation by systematically examining the relationship between surprisal and *n*-back performance at a more granular level.

Another important area for future research involves exploring potential age-related differences in task strategies within dual-task designs. We previously hypothesised that the differing effects of cognitive load on surprisal-driven reading times across age groups may reflect compensatory strategies in older adults. Given declines in executive control and working memory (*Idowu and Szameitat, 2023*; *Salthouse et al., 2003*; *Verhaeghen et al., 2003*), older adults may prioritise language processing over multitasking under high load, whereas younger adults might distribute resources more flexibly. Future studies should thus examine how age affects our response strategies in dual tasks.

Finally, it is important to note that GPT-2 next-word predictions are not tailored to individual participants. Large language models like GPT-2 are trained on extensive internet corpora (*Radford et al., 2019*). Consequently, this raises the question of whether linguistic content from the internet – and, by extension, GPT-2's language generation – disproportionately reflects the linguistic patterns of specific demographic or cognitive groups (*Gustilo and Dino, 2017*; *Haller et al., 2024*; *Hardy and Friginal, 2012*; *Tan and Celis, 2019*; *Venkit et al., 2023*). There is only very limited evidence addressing these questions directly. However, it has been shown that GPT-3.5 – a later-generation model building on the same transformer architecture as GPT-2 – shows performance comparable to children aged 6–15 in a language task requiring recollection and inference abilities (*Sicilia et al., 2023*). Additionally, another study shows that predictions generated by a German *GPT-2 Large* model (which we use in our studies as the smaller version, *GPT-2 Small*) align more closely with the language patterns of individuals with low verbal intelligence (*Haller et al., 2024*), complementing the findings by *Sicilia et al., 2023*. Despite the limited number of studies in this area, the findings from both studies suggest that the language predictions produced by GPT-2 may reflect certain demographic or cognitive traits, such as a very young age or a low verbal intelligence. As a result, language predictions generated by GPT-2 may not be equally representative across different groups, which must be considered when interpreting between-group effects in studies that rely on GPT-2-generated stimuli. In our study, this potential bias is unlikely to affect the results. We did not compare adults with children, and in the lab sample where verbal intelligence was measured ($N = 90$), it was only mildly correlated with age ($\rho = 0.279$, $p = 0.007$). Including verbal intelligence in our models did not improve model fits, and all main effects and interactions of age, cognitive load, and surprisal remained robust. Nevertheless, developing methods to generate individualised next-word predictions is an interesting direction for future research.

## Conclusion

In summary, the present study contributes to resolving the debate about the cognitive cost of predictive language processing. The data offer the following key insights:

First, low language predictability as well as high task demands both have a detrimental effect on reading time. This holds true across a large age range. Second, we find that higher task demands diminish the effects of language predictability on reading time, replicably demonstrating that language prediction draws on resources of executive control. Third, the data reveal age-related differences in the use of linguistic predictions: High predictability had more leverage on reading efficiency in older than in younger adults but was also more sensitive to available executive resources.

## Materials and methods

### Participants

We recorded data from 178 participants, who were either tested online (*N* = 83) or in a controlled lab environment (*N* = 95). We excluded data from three participants from the online sample from further analysis, either due to technical issues (*N* = 1) or because they reported having been distracted during the task (*N* = 2). The resulting final sample comprised 175 participants aged 18–85 years (*M* = 44.9 ± 17.9, 18–85 years) with a balanced gender distribution of 51% female, 47% male, and 2% non-binary identifying participants. All participants were native German speakers with normal or corrected-to-normal vision, and intact colour vision (assessed by a screening test; *Ishihara, 1987*). Exclusion criteria were a history of psychiatric or neurological disorders, drug abuse, dyslexia, illiteracy, or any impairments in language processing. Individuals who had consumed drugs or alcohol immediately prior to the study were not eligible for participation.

All participants from the online sample enlisted through the recruitment platform *Prolific*, whereas lab participants were recruited via an existing database of the Max-Planck-Institute. Lab participants above the age of 40 performed the Mini-Mental State Examination (*Folstein et al., 2014*) to screen for cognitive impairments. Middle-aged participants (40–59 years) had a mean score of 29.57 ± 0.84, whereas older participants (60–85 years) scored 28.23 ± 1.62 points.

### Study design

During the experiment, participants were asked to read short newspaper articles which were adapted from articles from the news archive of a well-known German magazine (*Der Spiegel*). The nine selected articles were edited to be easy to understand, neutral in tone and non-emotional in content to avoid any influences of text content on reading time. Additionally, all texts were limited to a length of 300 words (trials). Participants were instructed to read the texts carefully, as content-related multiple-choice questions were asked after each one. The comprehension questions served both as a measure of reading comprehension as well as a motivator to pay close attention to the reading material.

During this self-paced reading task (Reading Only task), each word was presented individually on screen and participants proceeded to the next word by pressing the space bar on their keyboard (*Figure 2a*). To ensure that each word was displayed at least briefly, the response window started after the word had been shown for a fixed period of 50 ms. Each word was presented centred on screen in one of four font colours (Hex codes: #D292F3 [lilac], #F989A2 [muted pink], #2AB7EF [cerulean blue], and #88BA3F [leaf green]) against a white background.

In four of the six main blocks, the self-paced reading task was complemented by a secondary *n*-back task (see *Figure 2a*), in which participants were instructed to press a target button on the keyboard ('C' for right-handed, 'M' for left-handed participants) whenever the font colour of the current word matched that of the previous (1-back Dual Task) or the penultimate word (2-back Dual Task). Participants were still required to press the spacebar to advance to the next word after pressing the target button. Reaction times were recorded for both kinds of responses. Neither the Reading Only task nor the 1-back and 2-back Dual Task blocks were speeded, allowing participants to complete the experiment at their own pace.

Before being introduced to this combination of reading and *n*-back task (see *Figure 2a*), participants could familiarise themselves with the *n*-back paradigm in short, non-linguistic, single-task blocks comprising coloured rectangles as stimuli. These blocks served as an introduction to the nature of an n-back task, and to quantify participants' working memory abilities. For all main blocks with an *n*-back task, a target ratio of 16.667% was used. The low target ratio was chosen to prevent an excessive number of *n*-back reactions during the dual-task blocks.

Taken together, the experiment comprised three cognitive load conditions: A baseline condition, comprising a reading task without additional *n*-back task (Reading Only), a reading task with an additional 1-back task (1-back Dual Task), and a reading task with an additional 2-back task (2-back Dual Task). Each condition was presented in two blocks, each comprising 300 trials. For each block, one of nine texts was randomly selected, with no text occurring more than once.

All experiments were implemented using *lab.js* (*Henninger et al., 2024*). Online studies were hosted on *OpenLab* – a server-side platform designed specifically for *lab.js* experiments (*Shevchenko, 2022*) – and data were saved on *OSF* (*Foster and Deardorff, 2017*).

## Generation of word surprisal and entropy scores

The predictability of each word was operationalised via word surprisal (*Figure 2b*), which reflects the predictability of the current word given its preceding context (*Smith and Levy, 2013*; *Hale, 2001*; *Levy, 2008*). A word's surprisal score is defined as the negative logarithm of the word's probability given its context (*Hale, 2001*):

$$surprisal_{w_n} = -\log\left(P\left(w_n \mid w_1, \ldots, w_{n-1}\right)\right)$$

If a word $w_n$ has a high surprisal score, its occurrence given its preceding context $w_1$, $w_2$, …, $w_{n-1}$ has a low probability, rendering it highly unpredictable (i.e., surprising).

In addition to each word's surprisal, we also computed the entropy of the probability distribution for each predicted word given its context (*Figure 2b*), which reflects the uncertainty in predicting the next word. The entropy is defined as the negative sum of the product of the probability of each word in the vocabulary and its respective logarithm probability (*Shannon, 1948*), or – put simply – as the average surprisal of all possible continuations in the vocabulary (*Slaats and Martin, 2023*):

$$entropy\left(p\right) = -\sum_{w_n \in W} P\left(w_n \mid w_1, \ldots, w_{n-1}\right) \log P\left(w_n \mid w_1, \ldots, w_{n-1}\right)$$

If entropy is low, only one or a few possible words in the vocabulary are assigned high probabilities of being the actual next word, hence indicating low uncertainty about which word will come next. This is usually the case if the previous context is very restricting. Conversely, if a vast amount of words in the vocabulary would be suited as continuations for the given context, and the probability distribution across words is fairly uniform, word entropy – and thus uncertainty about which word will come next – is high. Taken together, word surprisal signifies the predictability of each word whereas word entropy signifies the uncertainty underlying its prediction.

In the current study, we computed surprisal scores as well as one entropy score for each word in the experimental texts (mean surprisal: 18.165 ± 7.523, mean entropy: 4.067 ± 0.932; see further information in the *Appendix*).

Entropy and surprisal scores were estimated using a two-word context window. While short contexts have been shown to enhance GPT-2's psychometric alignment with human predictions, making next-word predictions more human-like (*Kuribayashi et al., 2022*), other work suggests that longer contexts can also increase model–human similarity (*Goldstein et al., 2022*). To reconcile these findings in our stimuli and guide the choice of context length, we tested longer windows and found surprisal scores were highly correlated with the 2-word context (e.g., 10-word vs. 2-word context: Spearman's $\rho$ = 0.976), with the overall pattern of results unchanged. Additionally, employing longer context windows would have also reduced the number of analysable trials, since surprisal cannot be computed for the first $k$ words of a text with a $k$-word context window. Crucially, any additional noise introduced by the short context biases effect estimates towards zero, making our analyses conservative rather than inflating them.

To generate word entropy and surprisal scores, we used a 12-layered GPT-2 model (*Radford et al., 2019*), which was pre-trained on German texts by the MDZ Digital Library team (dbmdz) at the Bavarian State Library, and the corresponding tokeniser, both available from the Hugging Face model hub (*Schweter, 2021*). Scores were calculated using Python version 3.10.12 (*Rossum and Drake, 2009*).

## Analysis

### Preprocessing

To gauge participants' response accuracy in the $n$-back task, we computed the detection-prime ($d'$) index. This measure quantifies the ability to distinguish between target and non-target stimuli, in our case trials (i.e., words in dual- and rectangles in single-task blocks) with colour repetitions and trials where the current colour does not match the colour from the nth previous trial. A $d'$ value of 0 signifies an inability to discriminate between signal and noise stimuli, suggesting that participants indicated they saw a target in either no or all trials. Thus, we excluded all dual-task blocks with $d$-primes of 0 from further analyses, which affected only five participants. In total, we excluded three main blocks in the online sample and two main blocks in the lab sample.

After each text block, participants were asked to answer three multiple-choice questions as a measure of reading comprehension. For each question, we showed four response options, with only one of them being correct. To ensure participants performed the tasks as intended and read the words on screen, we excluded all blocks where none of the questions were answered correctly. In total, we excluded one 1-back and seven 2-back Dual Task blocks from the online sample, as well as four 1-back and eleven 2-back Dual Task blocks from the lab sample, from datasets of 6 and 14 participants, respectively.

Lastly, we preprocessed the reading time data: First, any trial exhibiting a raw reading time exceeding 5000 ms was considered an extended break and subsequently excluded from further analyses. This cutoff was selected arbitrarily, based on the observation that participants tested in the lab did not exhibit trial durations exceeding 5000 ms. Therefore, we assume that participants tested online may have been distracted and less focused on the experiment during trial durations of this length.

To further remove outliers, we followed the procedure recommended by *Berger and Kiefer, 2021* to ensure comparable exclusion criteria for long and short outliers in typically skewed reading time: First, reading times were transformed using the *Proportion of Maximum Scaling* method (POMS; *Little, 2013*). We POMS-transformed the data on block level to account for potential differences in reading time distributions between blocks. The square root of each value was then taken to ensure a symmetric distribution. Following this, we *z*-transformed the data and excluded all trials from further analysis where *z*-scores fell outside a range of –2 to 2 (*Berger and Kiefer, 2021*; *Cousineau and Chartier, 2010*). Taken together, we excluded 12,968 trials (4.117% of all trials from the main blocks) with an average of 74.103 ± 12.654 excluded trials per participant.

To facilitate interpretability of units in the results, we subsequently continued working with the raw reading times, which had been cleaned of outliers at this stage, and log-transformed them for statistical analysis.

## Statistical analysis of *n*-back responses and comprehension questions

To ensure the validity of our cognitive load manipulation in the dual-task blocks, we examined whether increased cognitive load induced a decline both in *n*-back task performance – as indicated by reduced *d*-primes – and the accuracy in the comprehension question task, as reflected by a lower number of correct answers. We employed a linear mixed-effects model (LMM) for d-primes and a logistic linear mixed-effects model (GLMM, logit link function) for comprehension question accuracy.

In both models, we included recording location (online vs. lab), cognitive load (1-back and 2-back Dual Task vs. Reading Only as the reference level) and continuously measured age (centred) in both models as well as the interaction of age and cognitive load as fixed effects. In the model for the d-primes, we additionally included measures of comprehension question accuracy (on participant and block level) as well as the block number as fixed effects to control for different response strategies and tiredness effects, respectively. Moreover, we included the mean d-primes from the 1-back and 2-back Single Tasks as a working memory measure.

Please note that we did not control for trial-level stimulus colour here. The *n*-back task, which required participants to judge colours, was administered at the single-trial level, with colours pseudo-randomised to prevent any specific colour – or sequence of colours – from occurring more frequently than others. In contrast, comprehension questions were presented at the end of each block, meaning that trial-level stimulus colour was unrelated to accuracy on the block-level comprehension questions.

We assigned simple coding schemes to the factors recording location and cognitive load. While the model for *d*-primes included by-participant random slopes for cognitive load, the model for comprehension question accuracy comprised random intercepts for participants. Both models included random intercepts for participants and texts.

```
d-prime ~ mean d-prime from single tasks +
mean comprehension question accuracy +
block-level deviation from mean comprehension question accuracy +
recording location + block number +
age * cognitive load +
(1 + cognitive load | ID) +
```

```
(1 | text number)
```

Note. Structure of the model for *d*-primes in the *n*-back task in dual-task blocks. The variable age was centred and the variable cognitive load encompassed only two levels (1-back and 2-back) as there was no n-back task in the Reading Only condition.

```
comprehension question accuracy ~ recording location +
age * cognitive load +
(1 | ID) + (1 | text number)
```

Note. Structure of the model for accuracy in the comprehension question task. The variable age was centred. We used a binomial family distribution with a logit link function for modelling the comprehension question accuracies.

## Statistical analysis of reading times

We explored the effects of cognitive load, age, and surprise as well as their two- and three-way interactions on log-transformed reading times using an LMM. The model included an interaction of age, surprise, and cognitive load motivated by our hypotheses as well as additional fixed effects to control for nuisance effects. The final selection of fixed and random effects structure was based on highest $R^2$ values. As we only modelled reading times from trials where we had surprise scores, the first two trials of each block were not included in the statistical analyses.

We included the reading time of the previous word as a fixed effect to control for potential nuisance effects such as post-error slowing following a missed *n*-back target in the previous trial, or sequential modulation effects if the previous trial was ended prematurely, leading to an extended reading time carried over to the current trial. Additionally, it is important to consider that reading times, like many sequential behavioural measures, exhibit strong autocorrelation (*Schuckart et al., 2025*), meaning that the reading time of a given word is partially predictable from the reading time of the previous word. Such spillover effects can confound attempts to isolate trial-specific cognitive processes. As our primary goal was to model single-word prediction, we explicitly accounted for this autocorrelation by including the reading time of the preceding trial as a covariate.

As response strategies may differ between individuals, but also within an individual from block to block, we included two different regressors representing these distinct between- vs. within-participant effects on reading time. Between-participant effects were modelled by the individual mean comprehension performance whereas within-participant effects were modelled by the block-level deviation from this mean (cf. *Tune et al., 2021*; *Bell et al., 2019*). We further included block-wise d-primes and participant-wise mean single-task *d*-primes as a proxy of each participant's working memory capacity. By incorporating block- and participant-level performance measures, which are designed to be sensitive to task difficulty, we accounted for the potential variation in perceived task load between age groups or samples. For instance, the 2-back task might present a greater challenge for an older individual compared to their younger counterpart, therefore rendering the tasks not entirely comparable between age groups if not appropriately controlled for.

The remaining fixed effects entailed the recording location (online vs. lab), word frequency (as estimated using Python's wordfreq package; *Speer et al., 2018*), word length, as well as the position of block and trial in the course of the experiment as main effects. Furthermore, we included entropy as a fixed effect to account for the uncertainty in the prediction of the next word. Surprise and entropy values were weakly correlated ($r = 0.29$, $p < 0.001$). To account for the delay in reaction time associated with *n*-back responses, we included *n*-back reaction (reaction vs. no reaction) as a binary predictor in our models.

Finally, we included the three-way interaction of age (as a continuous predictor), cognitive load (on three levels: Reading Only, 1-back Dual Task, and 2-back Dual Task; contrasted using a simple coding scheme with the Reading Only condition as the reference level), and surprise score (continuous predictor). This entails the implicit inclusion of all two-way interactions of age, cognitive load, and surprise, as well as the main effects of those variables.

Random effects included random intercepts for participants, the effect of text, the current word, the colour of the current word, and by-participant random slopes for cognitive load. All continuous predictors were centred.

```
log(RT)~RT of previous word +
block-level d-prime+mean d-prime from single tasks +
mean comprehension question performance +
block-level deviation from mean comprehension question performance +
recording location +entropy +
word frequency +word length (without punctuation)+
n back reaction +block number +trial number +
surprisal * age * cognitive load +
(1+cognitive load | ID) + (1 | text number) + (1 | word) + (1 | colour)
```

Note. Model structure. RT = Reading Time, ID = participant.

To gain a more nuanced understanding of the three-way interaction of age, cognitive load, and surprisal, we performed a subsequent simple slopes analysis. This analysis allows exploring the interaction of two continuous predictors, in our case quantifying the slope of the surprisal effect in each of the three cognitive load conditions as a function of age. This way, we determined for which age range and cognitive load condition we observed a significant effect of surprisal on reading time.

For all models, including control analyses and the internal replication, *p*-values were obtained using ANOVAs with type III sums of squares. Degrees of freedom for *p*-values and standard errors were estimated using Satterthwaite's approximation for all LMMs, and Wald's approximation for the GLMM and GAM (*Luke, 2017*; *Satterthwaite, 1946*). All effects reported are significant on an alpha-level of 0.05 after FDR-correction for multiple comparisons (*Benjamini and Hochberg, 1995*).

All analyses were carried out in *R* version 4.2.2 (*R Development Core Team, 2023*) using the packages *gratia*, *interactions*, *lmerTest*, *lme4*, *mgcv*, *modelbased*, and *sjPlot* (*Gratia, 2024*; *Bates et al., 2015*; *Kuznetsova et al., 2017*; *Wood, 2011*; *Makowski et al., 2020*; *Long, 2024*).

## Control analysis: Dissociating cognitive control from attention

To disentangle attentional and cognitive load effects, we modelled reading times using an additional linear mixed model of the same structure as described before, but contrasting only the 2-back Dual Task condition with the less demanding 1-back Dual Task condition. The two Dual Task conditions only differ in cognitive load, but not attentional switching costs, which means any effects of cognitive load can be attributed to the cognitive load manipulation, with attentional demands held constant.

## Control analysis: Assessing potential effects of verbal intelligence and education

To ensure potential effects of verbal intelligence or education did not unduly influence our findings, we analysed data from 95 lab participants who reported their formal education in years (*M* = 18.0 ± 3.343, range = 11–30) and completed a lexical decision task – the Spot-the-Word test (*Baddeley et al., 1993*) – where they were asked to identify the word in pairs of words and non-words. Each participant's score on this test (*M* = 32.021 ± 2.993, range = 21–37) provided a measure of their verbal intelligence.

To assess the potential effect of education and verbal intelligence on reading time, we fitted three additional LMMs: The first model mirrored the structure of the original LMM used to analyse log-transformed reading times (see section *Statistical analysis of reading times* for the model structure), with one key modification: The predictor for recording location was excluded, as all participants were tested in a single location. The remaining two models followed the same structure, with the inclusion of centred education scores and centred Spot-the-Word test scores as additional predictors to account for education and verbal intelligence, respectively. We then statistically compared the baseline LMM with the two other LMMs using an ANOVA to determine whether verbal intelligence or education significantly improved the model fits.

## Control analysis: Modelling non-linear effects of age

We also fitted a generalised additive model (GAM) to our data to allow for non-linear relationships of the predictor variables with reading time, as it has been shown that reaction time is oftentimes modulated by predictor variables in a non-linear way (*Miwa and Baayen, 2021*; *Wood, 2017*).

Specifically, we employed a GAM to account for a potential non-linear relationship of age and surprisal with reading time. To this end, all continuous predictors were fitted with thin-plate regression splines and the interaction of surprisal, age, and cognitive load was fitted via a tensor product smooth with individual curves for each level of cognitive load. The number of basis dimensions for each smoothing spline was checked via model diagnostics available in *mgcv* after the first model set-up and appropriately updated to reach a *k*-index >1.01 and *p*>0.05 to avoid oversmoothing. The random effect structure was set up similarly to the LMM.

It is important to note that the outcomes of GAMs and LMMs can differ: GAMs are particularly adept at identifying localised, non-linear changes in predictor effects on reading time that may be overlooked by LMMs. As a result, the effects obtained from LMMs and those derived from GAMs are based on distinct metrics, which complicates direct comparisons between the two approaches.

## Control analysis: Disentangling the effect of cognitive load on pre- and post-stimulus predictive processing

Predictive processing can be conceptualised in terms of two complementary information-theoretic constructs: surprisal and entropy. While surprisal captures the cognitive cost associated with the post-stimulus integration of a generated prediction and the actual percept, entropy reflects the uncertainty underlying a next-word prediction (*Smith and Levy, 2013*; *Hale, 2001*; *Levy, 2008*; *Pimentel et al., 2022*; *Shannon, 1948*; *Slaats and Martin, 2023*). As such, entropy mirrors the estimated surprisal, suggesting entropy and surprisal are somewhat related, but do not represent the same construct (*Pimentel et al., 2022*).

While the primary focus of the present study is on predictive processing as indexed by word surprisal, it is also possible to partially dissociate pre- and post-stimulus predictive mechanisms by examining entropy. The empirical literature on the effect of entropy on reading time is scarce and somewhat contradictory, with evidence for both facilitative and inhibitory effects, or even no effects of next-word entropy over and above surprisal (*Pimentel et al., 2022*; *Karimi et al., 2024*; *Schijndel and Schuler, 2017*; *Aurnhammer and Frank, 2019*; *Roark et al., 2009*). Accordingly, we sought to investigate the effect of increased entropy in our dataset, asking whether higher entropy is associated with longer reading times, and whether this relationship is modulated by cognitive load, analogous to the effects observed for surprisal.

To this end, we fitted a linear mixed-effects model structurally analogous to that used for surprisal, substituting surprisal with entropy. Consistent with our approach for surprisal, we also included a three-way interaction of entropy with age and cognitive load to capture potential age-related changes in the relationship between entropy and cognitive load.

```
log(RT)~RT of previous word +
 block-level d-prime+mean d-prime from single tasks +
 mean comprehension question performance +
 block-level deviation from mean comprehension question performance +
 recording location +surprisal +
 word frequency +word length (without punctuation) +
 n-back reaction +block number +trial number +
 entropy * age * cognitive load +
 (1+cognitive load | ID) + (1 | text number) + (1 | word) + (1 | colour)
```

Note. Model structure. RT = Reading Time, ID = participant.

## Internal replication

To ensure the reliability of our findings, we conducted an internal replication of the previously described experiment. This internal replication was preregistered on OSF (doi: 10.17605/OSF.IO/SU6VX).

### Replication sample

As outlined in the preregistration, we conducted an online study with a sample of 100 participants. We excluded data from four participants from further analysis, either due to technical issues ($N = 2$) or because they reported having been distracted during the task ($N = 2$). The resulting final sample comprised 96 participants aged 18–70 years ($M = 39.750 \pm 13.996$ years) with a balanced gender distribution of 51% female, 48% male, and 1% non-binary identifying participants. As in the original experiment, all participants were native German speakers with normal or corrected-to-normal vision and intact colour vision without dyslexia, illiteracy, a history of psychiatric or neurological disorders or drug abuse. Individuals who had consumed drugs or alcohol immediately prior to the study were not eligible for participation.

### Replication analyses

Analogous to the original experiment, we first cleaned the reading time data of trials exceeding a duration of 5000 ms as well as outliers (see section *Preprocessing*), which affected 4.036% of all trials from the main blocks with an average of $72.656 \pm 11.089$ excluded trials per participant. The structure of the statistical model for the replication analysis was analogous to the model for the original analysis of reading times (see section *Statistical analysis of reading times*), except for the predictor of recording location, which was excluded as we only analysed data collected online. In addition, we modelled the two-way interaction between cognitive load and surprisal rather than the higher-order three-way interaction with age. This more parsimonious model structure was chosen to ensure adequate statistical power and to yield stable, interpretable estimates given the available sample sizes.

To compare results from the original experiment and the replication, we fitted the model once using the data of the online sample from the original experiment and once using the new online replication datasets.

Given that we simplified the analysis approach in the original study after having preregistered the replication, we deviated from the analysis plan described in the preregistration and made the same modifications here, resulting in the use of word surprisal for only one context length instead of four and, consequently, only one LMM instead of several.

## Acknowledgements

We thank Christian Koblitz and Marcel Blumenthal for their help with data acquisition in the lab. This work was supported by the German Research Foundation (DFG, OB 352/2-2 to JO, and HA 6314/4-2 to GH). GH was supported by Lise Meitner Excellence funding from the Max Planck Society and the European Research Council (ERC-2021-COG 101043747). Open access funding provided by the Max Planck Society. We acknowledge financial support by Land Schleswig-Holstein within the funding programme Open Access Publikationsfond.

## Additional information

### Competing interests

Jonas Obleser: Reviewing editor, eLife. The other authors declare that no competing interests exist.

### Funding

| Funder | Grant reference number | Author |
| --- | --- | --- |
| DFG | OB 352/2-2 | Jonas Obleser |
| DFG | HA 6314/4-2; Research Unit 5429/1 (467143400): HA 6314/10-1 | Gesa Hartwigsen |
| ERC | ERC-2021-COG 101043747 | Gesa Hartwigsen |

The funders had no role in study design, data collection, and interpretation, or the decision to submit the work for publication.

## Author contributions
Merle Marie Schuckart, Conceptualization, Data curation, Formal analysis, Investigation, Visualization, Methodology, Writing – original draft, Project administration, Writing – review and editing, shared first authorship with Sandra Martin; Sandra Martin, Conceptualization, Data curation, Formal analysis, Investigation, Methodology, Writing – original draft, Project administration, Writing – review and editing, shared first authorship with Merle Schuckart; Sarah Tune, Formal analysis, Methodology, Writing – review and editing; Lea-Maria Schmitt, Methodology, Writing – review and editing; Gesa Hartwigsen, Conceptualization, Supervision, Funding acquisition, Writing – review and editing, shared last authorship with Jonas Obleser; Jonas Obleser, Conceptualization, Supervision, Funding acquisition, Methodology, Writing – review and editing, shared last authorship with Gesa Hartwigsen

## Author ORCIDs
Merle Marie Schuckart ⓘ https://orcid.org/0000-0002-7178-7360
Sandra Martin ⓘ https://orcid.org/0000-0002-6687-5278
Sarah Tune ⓘ https://orcid.org/0000-0001-9022-9965
Lea-Maria Schmitt ⓘ https://orcid.org/0000-0002-9356-2234
Gesa Hartwigsen ⓘ https://orcid.org/0000-0002-8084-1330
Jonas Obleser ⓘ https://orcid.org/0000-0002-7619-0459

## Ethics
The study was conducted in accordance with the Declaration of Helsinki and was approved by the local ethics committees of the University of Lübeck and Leipzig University, respectively. Prior to participation, participants provided their written informed consent to participate and to have their data published, and received financial compensation (12€/h).

Reviewer #1 (Public review): https://doi.org/10.7554/eLife.108176.3.sa1
Reviewer #2 (Public review): https://doi.org/10.7554/eLife.108176.3.sa2
Author response https://doi.org/10.7554/eLife.108176.3.sa3

# Additional files

## Supplementary files
MDAR checklist

## Data availability
Experimental and analysis scripts as well as preprocessed data are publicly available in the project's OSF repository: https://osf.io/2hczy/.

The following dataset was generated:

| Author(s) | Year | Dataset title | Dataset URL | Database and Identifier |
| --- | --- | --- | --- | --- |
| Schuckart MM, Martin S, Tune S, Schmitt LM, Hartwigsen G, Obleser J | 2025 | Executive Resources Shape the Impact of Language Predictability Across the Adult Lifespan | https://osf.io/2hczy | Open Science Framework, 2hczy |

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

# Appendix 1

## Supplementary methods

### Demographics

We recruited participants both using the participant database of the Max-Planck-Institute in Leipzig and the online participant recruitment platform *Prolific*. Due to the predominantly younger demographic on *Prolific*, the online samples comprised mainly younger and middle-aged participants, whereas the lab sample spanned a broader age range (see *Figure 2—figure supplement 1*).

## Study design

### Text stimuli

During the experiment, participants were asked to read short newspaper articles on emotionally neutral topics such as literature, history, geography, and biology. All texts were edited to be easy to understand without excessive simplification. By doing so, we aimed to keep the cognitive load in the reading task as low as possible, while still maintaining a balance between text clarity and the avoidance of consistently high word predictability.

All texts had a Wiener Sachtextformel (WSTF4; *Bamberger and Lesen, 1984*) score below or equal to 10, which corresponds to a reading level suitable for students below 10th grade (mean WSTF4 score: 7.9 ± 0.623). As the WSTF4 only measures the syntactic complexity of a text, participants rated the subjective text difficulty as well as their subjective interest for two of the texts presented to them. All texts used in this study yielded a mean difficulty rating of 21.983 ± 2.799 (on a scale from 0 to 100, with 100 being 'extremely difficult') and a mean interest rating of 70.684 ± 5.657 (on a scale from 0 to 100, with 100 being 'extremely interesting'), confirming that the texts used in this study were both easy to understand and interesting to read.

## Structure of the experiment

At the outset of the experiment, participants were first presented with a training block of the Reading Only condition to familiarise themselves with the task. This was followed by the first main block of the Reading Only task (300 trials). After this, in a short training block (20 trials, repeating it was optional) followed by a longer main block (60 trials in the online experiment, 90 trials in the lab experiment), the participant was introduced to either the 1-back or the 2-back task as a non-linguistic single task comprising coloured rectangles as stimuli. Which *n*-back task was introduced first was randomised. This was then followed by the first dual-task block (300 trials) where the previously practised *n*-back task was performed together with the reading task. After having completed the first dual-task block in one of the two *n*-back conditions, the participant was then introduced to the other *n*-back task in the same fashion as before. Having performed each of the conditions once, the participant was subsequently presented with three main blocks of each of the three conditions in random order (300 trials each). After each block comprising a reading task, participants were asked to answer three multiple-choice comprehension questions on the content of the text.

**Appendix 1—table 1.** Results from models for task performance measures (*N* = 175).

| | LMM for *d*-primes | | | | | | GLMM for comprehension question accuracy | | | | |
| --- | --- | --- | --- | --- | --- | --- | --- | --- | --- | --- | --- |
| | Estimate | Std. error | t | df | p | | OR | Std. error | z | p | |
| Mean *d*-prime single-tasks | 0.469 | 0.0549 | 8.550 | 166.51 | $2.294 \times 10^{-14}$ | * | | | | | |
| Mean comprehension question performance | 0.011 | 0.0036 | 3.053 | 171.653 | $3.943 \times 10^{-3}$ | * | | | | | |
| De-meaned comprehension question performance | −0.001 | 0.0012 | −0.475 | 372.688 | $6.350 \times 10^{-1}$ | | | | | | |
| Block number | −0.006 | 0.0085 | −0.676 | 349.333 | $5.618 \times 10^{-1}$ | | | | | | |
| Recording location [online] | −0.506 | 0.0902 | −5.609 | 163.182 | $1.920 \times 10^{-7}$ | * | 0.980 | 0.1876 | −0.106 | $9.15 \times 10^{-1}$ | |
| Age | −0.005 | 0.0027 | −2.057 | 164.038 | $5.003 \times 10^{-2}$ | | 0.986 | 0.0053 | −2.676 | $1.304 \times 10^{-2}$ | * |
| Cognitive load [1-back vs. Reading Only] | | | | | | | 0.253 | 0.0390 | −8.928 | $1.011 \times 10^{-18}$ | * |
| Cognitive load [2-back vs. Reading Only] | | | | | | | 0.156 | 0.0234 | −12.403 | $1.753 \times 10^{-34}$ | * |
| Cognitive load. [2-back vs. 1-back] | −1.636 | 0.0626 | −26.120 | 173.125 | $2.672 \times 10^{-61}$ | * | | | | | |
| Age * cognitive load [1-back vs. Reading Only] | | | | | | | 0.990 | 0.0081 | −1.183 | $3.313 \times 10^{-1}$ | |
| Age * cognitive load [2-back vs. Reading Only] | | | | | | | 1.003 | 0.0080 | 0.395 | $8.081 \times 10^{-1}$ | |
| Age * cognitive load [2-back vs. 1-back] | −0.014 | 0.0035 | −3.931 | 169.766 | $2.210 \times 10^{-4}$ | * | | | | | |
| | Conditional/marginal $R^2$ | | | | ICC | | Conditional/marginal $R^2$ | | | ICC | |
| Model fit | 0.822/0.634 | | | | 0.512 | | 0.304/0.146 | | | 0.185 | |

Note. All continuous predictors were centred. Degrees of freedom for p-values, standard errors and confidence intervals (CI) were computed using Satterthwaite's approximation (LMM for *d*-primes) and Wald's approximation (GLMM for comprehension question accuracy). All p-values reported here are FDR-corrected and were computed using ANOVAs with type III sums of squares. Results that are significant on an alpha-level of 0.05 are marked with a star. OR = Odds Ratio.

**Appendix 1—table 2.** Results from the model for reading times for full original sample (N = 175).

| | Predictors | Estimate | Std. error | CI | t | df | p | |
|---|---|---|---|---|---|---|---|---|
| | Reading time of previous trial (log-transformed) | 0.110235 | 0.001409 | 0.107472 to 0.112997 | 78.210829 | 287,711.33 | $<1.33 \times 10^{-322}$ | * |
| | d-prime | −0.006293 | 0.001767 | −0.009755 to −0.002831 | −3.562277 | 224,716.70 | $4.903 \times 10^{-4}$ | * |
| | Mean d-prime single-tasks | 0.084636 | 0.019585 | 0.045974 to 0.123298 | 4.321418 | 169.86 | $3.717 \times 10^{-5}$ | * |
| | Mean comprehension question performance | 0.002841 | 0.001276 | 0.000321 to 0.005360 | 2.225728 | 169.26 | $3.282 \times 10^{-2}$ | * |
| | De-meaned comprehension question performance | 0.000053 | 0.000038 | −0.000023 to 0.000128 | 1.374475 | 257,871.34 | $1.693 \times 10^{-1}$ | |
| | Word frequency | 0.643570 | 0.362057 | −0.067231 to 1.354371 | 1.777538 | 727.78 | $8.280 \times 10^{-2}$ | |
| | Word length | 0.007839 | 0.000407 | 0.007039 to 0.008638 | 19.240951 | 1400.80 | $7.407 \times 10^{-73}$ | * |
| | Word entropy | 0.001410 | 0.000785 | −0.000129 to 0.002949 | 1.796533 | 7594.27 | $8.280 \times 10^{-2}$ | |
| | n-back reaction [reaction vs. no reaction] | 0.317464 | 0.001799 | 0.313937 to 0.320990 | 176.444743 | 287,862.05 | $<1.33 \times 10^{-322}$ | * |
| | Block number | −0.006495 | 0.000164 | −0.006816 to −0.006173 | −39.637620 | 287,253.02 | $<1.33 \times 10^{-322}$ | * |
| | Trial number | −0.000456 | 0.000007 | −0.000470 to −0.000442 | −64.085235 | 18,673.68 | $<1.33 \times 10^{-322}$ | * |
| | Recording location [online vs. lab] | −0.219339 | 0.032284 | −0.283072 to −0.155606 | −6.793975 | 168.69 | $2.686 \times 10^{-10}$ | * |
| Main effects | Surprisal | 0.001707 | 0.000151 | 0.001411 to 0.002002 | 11.320677 | 2361.37 | $1.368 \times 10^{-28}$ | * |
| | Age | 0.009113 | 0.000991 | 0.007158 to 0.011068 | 9.199100 | 178.46 | $1.751 \times 10^{-16}$ | * |
| | Cognitive load [1-back vs. Reading Only] | 0.473800 | 0.013916 | 0.446336 to 0.501264 | 34.046321 | 176.18 | $8.399 \times 10^{-79}$ | * |
| | Cognitive load [2-back vs. Reading Only] | 0.791540 | 0.026090 | 0.740046 to 0.843034 | 30.338989 | 173.76 | $7.320 \times 10^{-71}$ | * |
| | Surprisal x age | 0.000035 | 0.000004 | 0.000027 to 0.000042 | 9.287151 | 287,771.27 | $3.481 \times 10^{-20}$ | * |
| | Surprisal x cognitive load [1-back vs. Reading Only] | −0.001093 | 0.000161 | −0.001409 to −0.000776 | −6.771521 | 287,959.11 | $2.043 \times 10^{-11}$ | * |
| | Surprisal x cognitive load [2-back vs. Reading Only] | −0.001255 | 0.000163 | −0.001575 to −0.000935 | −7.681261 | 288,294.96 | $2.709 \times 10^{-14}$ | * |
| | Age x cognitive load [1-back vs. Reading Only] | −0.002798 | 0.000776 | −0.004330 to −0.001267 | −3.606479 | 171.99 | $5.135 \times 10^{-4}$ | * |
| Two-way interactions | Age x cognitive load [2-back vs. Reading Only] | −0.002458 | 0.001454 | −0.005329 to 0.000412 | −1.690400 | 170.79 | $9.681 \times 10^{-2}$ | |

*Appendix 1—table 2 continued*

| | | LMM for full original sample (*N* = 175) | | | | | |
|---|---|---|---|---|---|---|---|
| **Three-way interactions** | Surprisal x age x cognitive load [1-back vs. Reading Only] | −0.000111 | 0.000009 | −0.000129 to −0.000094 | −12.266076 | 287,807.34 | $3.748\times10^{-34}$ | * |
| | Surprisal x age x cognitive load [2-back vs. Reading Only] | −0.000078 | 0.000009 | −0.000096 to −0.000060 | −8.483676 | 287,771.65 | $4.384\times10^{-17}$ | * |
| | Intra-class correlation (ICC) | 0.46 | | | | | |
| **Model fit** | Marginal $R^2$/conditional $R^2$ | 0.643/0.807 | | | | | |

Note. All continuous predictors were centred. Degrees of freedom for p-values, standard errors and confidence intervals (CI) were computed using Satterthwaite's approximation. All p-values reported here are FDR-corrected and were computed using ANOVAs with type III sum of squares. Results that are significant on an alpha-level of 0.05 are marked with a star.

**Appendix 1—table 3.** Results from models for reading times for original online sample and online replication sample (*N* = 80 and *N* = 96, respectively).

| Predictors | LMM for online original sample (N = 80) | | | | | | | LMM for online replication sample (N = 96) | | | | | | |
|---|---|---|---|---|---|---|---|---|---|---|---|---|---|---|
| | Estimate | Std. error | CI | t | df | p | | Estimate | Std. Error | CI | t | df | p | |
| Reading time of previous trial (log-transformed) | 0.076157 | 0.002060 | 0.072120 to 0.080193 | 36.978112 | 133,340.940 | $5.049 \times 10^{-297}$ | * | 0.149035 | 0.001818 | 0.145472 to 0.152597 | 81.985 | 161,495.845 | $<3.442 \times 10^{-281}$ | * |
| d-prime | 0.058086 | 0.003309 | 0.051601 to 0.064571 | 17.555499 | 58,135.052 | $2.440 \times 10^{-68}$ | * | -0.010767 | 0.002175 | -0.015029 to -0.006504 | -4.950607 | 139,351.725 | $8.888 \times 10^{-7}$ | * |
| Mean *d*-prime single-tasks | 0.093533 | 0.027543 | 0.038674 to 0.148391 | 3.395825 | 75.930447 | $1.403 \times 10^{-3}$ | * | 0.111742 | 0.020097 | 0.071832 to 0.151652 | 5.560231 | 92.614 | $3.324 \times 10^{-7}$ | * |
| Mean comprehension question performance | 0.003366 | 0.001733 | -0.000085 to 0.006816 | 1.942677 | 76.169 | $6.690 \times 10^{-2}$ | | 0.003459 | 0.001659 | 0.000163 to 0.006754 | 2.084567 | 91.938 | $4.487 \times 10^{-2}$ | * |
| De-meaned comprehension question performance | -0.000481 | 0.000059 | -0.000597 to -0.000365 | -8.123202 | 118,882.498 | $8.250 \times 10^{-16}$ | * | -0.000715 | 0.000047 | -0.000807 to -0.000624 | -15.286171 | 152,148.746 | $2.661 \times 10^{-52}$ | * |
| Word frequency | 0.255248 | 0.300091 | -0.335305 to 0.845800 | 0.850567 | 299.741 | $4.190 \times 10^{-1}$ | | 0.277510 | 0.253935 | -0.222530 to 0.777550 | 1.092838 | 258.972 | $2.917 \times 10^{-1}$ | |
| Word length | 0.006329 | 0.000414 | 0.005517 to 0.007141 | 15.297105 | 1292.834 | $2.874 \times 10^{-48}$ | * | 0.006341 | 0.000362 | 0.005631 to 0.007052 | 17.512071 | 1287.953 | $2.794 \times 10^{-61}$ | * |
| Word entropy | -0.000656 | 0.000933 | -0.002485 to 0.001173 | -0.702998 | 3838.651 | $4.821 \times 10^{-1}$ | | 0.000337 | 0.000826 | -0.001283 to 0.001957 | 0.407433 | 3601.355 | $6.837 \times 10^{-1}$ | |
| n-back reaction [reaction vs. no reaction] | 0.366351 | 0.002603 | 0.361250 to 0.371452 | 140.757916 | 133,345.032 | $<5.049 \times 10^{-297}$ | * | 0.335449 | 0.002293 | 0.330955 to 0.339943 | 146.303524 | 161,785.798 | $<3.442 \times 10^{-281}$ | * |
| Block number | -0.008486 | 0.000247 | -0.008970 to -0.008002 | -34.367684 | 131,940.893 | $4.797 \times 10^{-257}$ | * | -0.008042 | 0.000224 | -0.008481 to -0.007604 | -35.946825 | 159,855.434 | $3.442 \times 10^{-281}$ | * |
| Trial number | -0.000407 | 0.000009 | -0.000425 to -0.000390 | -45.720722 | 8566.066 | $<5.049 \times 10^{-297}$ | * | -0.000386 | 0.000008 | -0.000402 to -0.000371 | -48.601047 | 8175.168 | $<3.442 \times 10^{-281}$ | * |
| Surprisal | 0.001145 | 0.000162 | 0.000826 to 0.001463 | 7.046510 | 1889.625 | $4.190 \times 10^{-12}$ | * | 0.001375 | 0.000144 | 0.001093 to 0.001656 | 9.578258 | 1886.753 | $5.358 \times 10^{-21}$ | * |
| Age | 0.005382 | 0.001535 | 0.002326 to 0.008438 | 3.507190 | 76.056 | $1.057 \times 10^{-3}$ | * | 0.008953 | 0.001318 | 0.006336 to 0.011570 | 6.795181 | 91.979 | $1.460 \times 10^{-9}$ | * |
| Cognitive load [1-back vs. Reading Only] | 0.468989 | 0.018722 | 0.431749 to 0.506229 | 25.050620 | 82.526 | $5.650 \times 10^{-40}$ | * | 0.507557 | 0.022113 | 0.463669 to 0.551446 | 22.953170 | 96.841 | $1.343 \times 10^{-40}$ | * |
| Cognitive load [2-back vs. Reading Only] | 0.824086 | 0.034653 | 0.755102 to 0.893071 | 23.781266 | 78.282 | $2.909 \times 10^{-37}$ | * | 0.722423 | 0.031793 | 0.659316 to 0.785531 | 22.722673 | 96.183 | $3.806 \times 10^{-40}$ | * |

Main effects

*Appendix 1—table 3 continued on next page*

*Appendix 1—table 3 continued*

| | **LMM for online original sample (N = 80)** | | | | | | **LMM for online replication sample (N = 96)** | | | | | | |
|---|---|---|---|---|---|---|---|---|---|---|---|---|---|
| | Estimate | SE | CI | t | df | p | | Estimate | SE | CI | t | df | p |
| **Two-way interactions** | | | | | | | | | | | | | |
| Surprisal x cognitive load [1-back vs. Reading Only] | 0.000786 | 0.000223 | 0.000349 to 0.001223 | 3.522647 | 133,382.112 | **6.411×10⁻⁴** | * | 0.001499 | 0.000203 | 0.001101 to 0.001897 | 7.376731 | 161,262.305 | **2.667×10⁻¹³** * |
| Surprisal x cognitive load [2-back vs. Reading Only] | 0.000375 | 0.000225 | −0.000067 to 0.000816 | 1.661967 | 133,507.349 | 1.086×10⁻¹ | | 0.001365 | 0.000203 | 0.000967 to 0.001763 | 6.721136 | 161,923.055 | **2.714×10⁻¹¹** * |
| Intra-class correlation (ICC) | 0.47 | | | | | | | 0.50 | | | | | |
| **Model fit** | | | | | | | | | | | | | |
| Marginal R²/conditional R² | 0.587/0.781 | | | | | | | 0.615/0.809 | | | | | |

Note. All continuous predictors were centred. Degrees of freedom for p-values, standard errors and confidence intervals (CI) were computed using Satterthwaite's approximation. All p-values reported here are FDR-corrected and were computed using ANOVAs with type III sum of squares. Results that are significant on an alpha-level of 0.05 are marked with a star.

**Appendix 1—table 4.** Results from models for control analysis (1-back vs. 2-back) of reading times for full original sample (N = 175).

Control analysis 2-back vs. 1-back: LMM for full original sample (*N* = 175)

| | Predictors | Estimate | Std. error | CI | t | df | p | |
|---|---|---|---|---|---|---|---|---|
| | Reading time of previous trial (log-transformed) | 0.062181 | 0.001755 | 0.058740 to 0.065622 | 35.420630 | 188,475.92 | $3.296 \times 10^{-273}$ | * |
| | d-prime | −0.006667 | 0.001898 | −0.010388 to −0.002947 | −3.512595 | 168,020.99 | $7.398 \times 10^{-4}$ | * |
| | Mean *d*-prime single-tasks | 0.094749 | 0.020772 | 0.053746 to 0.135752 | 4.561318 | 171.05 | $1.756 \times 10^{-5}$ | * |
| | Mean comprehension question performance | 0.003146 | 0.001352 | 0.000478 to 0.005815 | 2.327238 | 170.38 | $3.018 \times 10^{-2}$ | * |
| | De-meaned comprehension question performance | 0.000038 | 0.000047 | −0.000053 to 0.00013 | 0.821932 | 150,546.36 | $4.837 \times 10^{-1}$ | |
| | Word frequency | −0.128276 | 0.319365 | −0.756126 to 0.499574 | −0.401659 | 398.70 | $6.882 \times 10^{-1}$ | |
| | Word length | 0.005220 | 0.000414 | 0.004408 to 0.006031 | 12.616091 | 1351.55 | $4.020 \times 10^{-34}$ | * |
| | Word entropy | 0.001711 | 0.000914 | −0.000081 to 0.003502 | 1.871897 | 4521.92 | $8.171 \times 10^{-2}$ | |
| | n-back reaction [reaction vs. no reaction] | 0.316007 | 0.001917 | 0.312250 to 0.319764 | 164.836212 | 188,099.24 | $<1.33 \times 10^{-322}$ | * |
| | Block number | −0.011363 | 0.000295 | −0.011941 to −0.01079 | −38.535522 | 185,049.49 | $1.33 \times 10^{-322}$ | * |
| | Trial number | −0.000450 | 0.000009 | −0.000467 to −0.000433 | −52.053811 | 10,059.10 | $<1.33 \times 10^{-322}$ | * |
| | Recording location [online vs. lab] | −0.226062 | 0.034131 | −0.293438 to −0.158686 | −6.623323 | 169.71 | $8.894 \times 10^{-10}$ | * |
| | Surprisal | 0.001848 | 0.000161 | 0.001532 to 0.002164 | 11.467327 | 2010.37 | $3.901 \times 10^{-29}$ | * |
| | Age | 0.008762 | 0.001101 | 0.006591 to 0.010934 | 7.961804 | 184.31 | $3.751 \times 10^{-13}$ | * |
| Main effects | Cognitive load [2-back vs. 1-back] | 0.338896 | 0.020892 | 0.297669 to 0.380123 | 16.221045 | 178.98 | $1.838 \times 10^{-36}$ | * |
| | Surprisal x age | 0.000003 | 0.000005 | −0.000007 to 0.000012 | 0.545912 | 187,940.82 | $6.159 \times 10^{-1}$ | |
| | Surprisal x cognitive load [2-back vs. 1-back] | −0.000148 | 0.000173 | −0.000486 to 0.000191 | −0.855146 | 187,910.28 | $4.837 \times 10^{-1}$ | |
| Two-way interactions | Age x cognitive load [2-back vs. 1-back] | 0.000689 | 0.001156 | −0.001593 to 0.00297 | 0.595977 | 172.03 | $6.133 \times 10^{-1}$ | |
| Three-way interaction | Surprisal x age x cognitive load [2-back vs. 1-back] | 0.000033 | 0.000010 | 0.000014 to 0.000052 | 3.372931 | 188,203.53 | $1.144 \times 10^{-3}$ | * |
| | Intra-class correlation (ICC) | 0.44 | | | | | | |
| Model fit | Marginal $R^2$/conditional $R^2$ | 0.442/0.690 | | | | | | |

*Appendix 1—table 4 continued on next page*

*Appendix 1—table 4 continued*

**Control analysis 2-back vs. 1-back: LMM for full original sample (*N* = 175)**

*Note.* p-values were computed using Wald's approximation as implemented in the package mgcv. Results that are significant on an alpha-level of 0.05 are marked with a star. Edf: Effective degrees of freedom.

**Appendix 1—table 5.** Results from GAM for control analysis of reading times for full original sample (*N* = 175).

**Control analysis: GAM for full original sample (*N* = 175)**

| Predictors | Estimate | Std. Error | t | F | EDF | p | |
|---|---|---|---|---|---|---|---|
| Reading time of previous trial (log-transformed) | | | | 204.591 | 33.664 | <2 × 10⁻¹⁶ | * |
| *d*-prime | | | | 38.774 | 28.042 | <2 × 10⁻¹⁶ | * |
| Mean *d*-prime single-tasks | | | | 24.344 | 1.892 | <2 × 10⁻¹⁶ | * |
| Mean comprehension question performance | | | | 5.348 | 2.305 | 3.23×10⁻³ | * |
| De-meaned comprehension question performance | | | | 39.408 | 7.477 | <2 × 10⁻¹⁶ | * |
| Word frequency | | | | 6.837 | 7.571 | <2 × 10⁻¹⁶ | * |
| Word length | | | | 73.076 | 4.038 | <2 × 10⁻¹⁶ | * |
| Word entropy | | | | 4.027 | 3.704 | 2.44×10⁻³ | * |
| Surprisal | | | | 9.547 | 4.107 | <2 × 10⁻¹⁶ | * |
| Age | | | | 51.783 | 3.028 | <2 × 10⁻¹⁶ | * |
| n-back reaction [reaction vs. no reaction] | 0.3167 | 0.00179 | 177.06 | | | <2 × 10⁻¹⁶ | * |
| Block number | −0.0061 | 0.00017 | −36.89 | | | <2 × 10⁻¹⁶ | * |
| Trial number | −0.0004 | 0.00001 | −64.47 | | | <2 × 10⁻¹⁶ | * |
| Recording location (online vs. lab) | −0.2514 | 0.02602 | −9.66 | | | <2 × 10⁻¹⁶ | * |
| Cognitive load [1-back vs. Reading Only] | 0.4318 | 0.02514 | 17.17 | | | <2 × 10⁻¹⁶ | * |
| Cognitive load [2-back vs. Reading Only] | 0.7819 | 0.02526 | 30.95 | | | <2 × 10⁻¹⁶ | * |
| **Main effects** | | | | | | | |
| **Two-way interactions** | | | | | | | |
| Surprisal x cognitive load | | | | 13.962 | 13.849 | <2 × 10⁻¹⁶ | * |
| Sprisal x age x cognitive load [Reading Only] | | | | 23.946 | 10.248 | <2 × 10⁻¹⁶ | * |
| Surprisal x age x cognitive load [1-back] | | | | 2.874 | 2.017 | 3.616×10⁻² | * |
| **Three-way interactions** | | | | | | | |
| Surprisal x age x cognitive load [2-back] | | | | 2.392 | 4.877 | 2.375×10⁻² | * |

*Appendix 1—table 5 continued on next page*

*Appendix 1—table 5 continued*

**Control analysis: GAM for full original sample (*N* = 175)**

| | | | | | |
|---|---|---|---|---|---|
| | Cognitive load \| ID | 255.250 | 508.610 | $<2 \times 10^{-16}$ | * |
| | Text Nr. | 14,053.220 | 7.840 | $<2 \times 10^{-16}$ | * |
| | Word | 1.870 | 804.600 | $<2 \times 10^{-16}$ | * |
| **Random effects** | Colour | 12.530 | 2.770 | $<2 \times 10^{-16}$ | * |
| **Model fit** | $R^2$ | 815 | | | |

*Note.* All continuous predictors were centred. Degrees of freedom for *p*-values, standard errors and confidence intervals (CI) were computed using Satterthwaite's approximation. All *p*-values reported here are FDR-corrected and were computed using ANOVAs with type III sum of squares. Results that are significant on an alpha-level of 0.05 are marked with a star.

**Appendix 1—table 6.** Results from the model for reading times for full original sample (N = 175) for the effects of entropy, cognitive load, and age on reading time.

**LMM for full original sample (N = 175)**

| Predictors | Estimate | Std. error | CI | t | df | p | |
|---|---|---|---|---|---|---|---|
| Reading time of previous trial (log-transformed) | 0.110066 | 0.001410 | 0.107302 to 0.112830 | 78.048894 | 287,686.846 | $<8.542 \times 10^{-79}$ | * |
| d-prime | −0.006248 | 0.001767 | −0.009712 to −0.002784 | −3.535537 | 224,708.066 | $6.105 \times 10^{-4}$ | * |
| Mean d-prime single-tasks | 0.084628 | 0.019587 | 0.045963 to 0.123292 | 4.320683 | 169.899 | $4.225 \times 10^{-5}$ | * |
| Mean comprehension question performance | 0.002842 | 0.001276 | 0.000322 to 0.005362 | 2.226655 | 169.295 | $3.275 \times 10^{-2}$ | * |
| De-meaned comprehension question performance | 0.000052 | 0.000039 | −0.000023 to 0.000127 | 1.350491 | 257,990.129 | $1.769 \times 10^{-1}$ | |
| Word frequency | 0.608299 | 0.360822 | −0.100080 to 1.316678 | 1.685873 | 725.615 | $9.929 \times 10^{-2}$ | |
| Word length | 0.007812 | 0.000406 | 0.007015 to 0.008609 | 19.220944 | 1402.270 | $9.864 \times 10^{-73}$ | * |
| Surprisal | 0.001682 | 0.000150 | 0.001387 to 0.001977 | 11.177024 | 2359.191 | $7.143 \times 10^{-28}$ | * |
| n-back reaction [reaction vs. no reaction] | 0.317360 | 0.001800 | 0.313832 to 0.320888 | 176.311387 | 287,866.290 | $<8.542 \times 10^{-79}$ | * |
| Block number | −0.006481 | 0.000164 | −0.006803 to −0.006160 | −39.539875 | 287,255.082 | $<8.542 \times 10^{-79}$ | * |
| Trial number | −0.000456 | 0.000007 | −0.000470 to −0.000442 | −64.065173 | 18,557.632 | $<8.542 \times 10^{-79}$ | * |
| Recording location [online vs. lab] | −0.219370 | 0.032287 | −0.283107 to −0.155632 | −6.794462 | 168.720 | $3.895 \times 10^{-10}$ | * |
| Entropy | 0.001412 | 0.000785 | −0.000126 to 0.002950 | 1.800222 | 7570.659 | $8.213 \times 10^{-2}$ | |
| Age | 0.009110 | 0.000991 | 0.007155 to 0.011065 | 9.195551 | 178.495 | $2.325 \times 10^{-16}$ | * |
| Cognitive load [1-back vs. Reading Only] | 0.473980 | 0.013924 | 0.446501 to 0.501458 | 34.041427 | 176.188 | $8.542 \times 10^{-79}$ | * |
| Main effects Cognitive load [2-back vs. Reading Only] | 0.791850 | 0.026098 | 0.740340 to 0.843361 | 30.341059 | 173.750 | $7.294 \times 10^{-71}$ | * |

*Appendix 1—table 6 continued on next page*

*Appendix 1—table 6 continued*

**LMM for full original sample (N = 175)**

| | | | | | | | |
|---|---|---|---|---|---|---|---|
| | Entropy x age | 0.000090 | 0.000030 | 0.000032 to 0.000148 | 3.030333 | 287,391.572 | **3.257×10$^{-3}$** | * |
| | Entropy x cognitive load [1-back vs. Reading Only] | 0.006638 | 0.001273 | 0.004142 to 0.009133 | 5.213908 | 287,500.035 | **3.416×10$^{-7}$** | * |
| | Entropy x cognitive load [2-back vs. Reading Only] | 0.006490 | 0.001291 | 0.003959 to 0.009021 | 5.025891 | 287,757.942 | **8.595×10$^{-7}$** | * |
| | Age x cognitive load [1-back vs. Reading Only] | −0.002785 | 0.000776 | −0.004317 to −0.001253 | −3.587447 | 171.994 | **6.142×10$^{-4}$** | * |
| Two-way interactions | Age x cognitive load [2-back vs. Reading Only] | −0.002441 | 0.001455 | −0.005313 to 0.000430 | −1.678106 | 170.772 | **9.929×10$^{-2}$** | |
| | Entropy x age x cognitive load [1-back vs. Reading Only] | −0.000399 | 0.000072 | −0.000540 to −0.000258 | −5.546582 | 287,440.357 | **5.831×10$^{-8}$** | * |
| Three-way interactions | Entropy x age x cognitive load [2-back vs. Reading Only] | −0.000188 | 0.000073 | −0.000331 to −0.000045 | −2.577310 | 287,488.317 | **1.258×10$^{-2}$** | * |
| | Intra-class correlation (ICC) | 0.46 | | | | | | |
| **Model fit** | Marginal $R^2$/conditional $R^2$ | 0.643/0.807 | | | | | | |

*Note.* All continuous predictors were centred. Degrees of freedom for *p*-values, standard errors and confidence intervals (CI) were computed using Satterthwaite's approximation. All *p*-values reported here are FDR-corrected and were computed using ANOVAs with type III sum of squares. Results that are significant on an alpha-level of 0.05 are marked with a star.

