## [Editor Report · eLife Assessment]

This study presents a **valuable** finding on whether executive resources mediate the impact of language predictability in reading in the context of aging. The evidence is **solid** in the investigation of prediction in reading, with one caveat that the text materials used could be biased against the aging population. The work will be of interest to cognitive neuroscientists working on reading, language comprehension, and executive control.

---

## [Referee Report · Reviewer #1 (Public review)]

The authors of this study set out to address a central question in the psycholinguistics literature: does the human brain's ability to predict upcoming language come at a cognitive cost, or is it an automatic, "free" process? To investigate this, they employed a dual-task paradigm where participants read texts word-by-word while simultaneously performing a secondary task (an n-back task on font color) designed to manipulate cognitive load. The study examines how this external cognitive load, along with the effects of aging, modulates the impact of word predictability (measured by surprisal and entropy) on reading times. The central finding is that increased cognitive load diminishes the effects of word predictability, supporting the conclusion that language prediction is a resource-dependent process.

A major strength of the revised manuscript is its comprehensive and parallel analysis of both word surprisal and entropy. The initial submission focused almost exclusively on surprisal, which primarily reflects the cost of integrating a word into its context after it has been perceived. The new analysis now thoroughly investigates entropy as well, which reflects the uncertainty and cognitive effort involved in predicting the next word before it appears. This addition provides a much more complete and theoretically nuanced picture, allowing the authors to address how cognitive load affects both predictive and integrative stages of language processing. This is a significant improvement and substantially increases the paper's contribution to the field.

Furthermore, the authors have commendably addressed the initial concerns regarding the robustness of their replication findings. The first version of the manuscript presented replication results that were inconsistent, particularly for key interaction effects. In the revision, the authors have adopted a more focused and appropriately powered modeling approach for the replication analysis. This revised analysis now demonstrates a consistent effect of cognitive load on the processing of predictable words across both the original and replication datasets. This strengthens the evidence for the paper's primary claim.

The initial review also raised concerns that the results could be explained by general cognitive factors, such as task-switching costs, rather than the specific demands on the language prediction system. While the complexity of cognitive load in a dual-task paradigm remains a challenge, the authors have provided sufficient justification in their revisions and rebuttal to support their interpretation that the observed effects are genuinely tied to the process of language prediction.

---

## [Referee Report · Reviewer #2 (Public review)]

Summary:

This paper considers the effects of cognitive load (using an n-back task related to font color), predictability, and age on reading times in two experiments. There were main effects of all predictors, but more interesting effects of load and age on predictability. The effect of load is very interesting, but the manipulation of age is problematic, because we don't know what is predictable for different participants (in relation to their age). There are some theoretical concerns about prediction and predictability, and a need to address literature (reading time, visual world, ERP studies).

There is a major concern about the effects of age. See the results (155-190): this depends what is meant by word predictability. It's correct if it means the predictability in the corpus. But it may or may not be correct if it refers to how predictable a word is to an individual participant. The texts are unlikely to be equally predictable to different participants, and in particular to younger vs. older participants, because of their different experience. To put it informally, the newspaper articles may be more geared to the expectations of younger people. But there is also another problem: the LLM may have learned on the basis of language that has largely been produced by young people and so its predictions are based on what young people are likely to say. Both of these possibilities strike me as extremely likely. So it may be that older adults are affected more by words that they find surprising, but it is also possible that the texts are not what they expect, or the LLM predictions from the text are not the ones that they would make. In sum, I am not convinced that the authors can say anything about the effects of age unless they can determine what is predictable for different ages of participants. I suspect that this failure to control is an endemic problem in the literature on aging and language processing and needs to be systematically addressed.

Overall, I think the paper makes enough of a contribution with respect to load to be useful to the literature. But for discussion of age, we would need something like evidence of how younger and older adults would complete these texts (on a word-by-word basis) and that they were equally predictable for different ages. I assume there are ways to get LLMs to emulate different participant groups, but I doubt if we could be confident about their accuracy without a lot of testing. But without something like this, I think making claims about age would be quite misleading.

The authors respond to my summary comment by saying that prediction is individual and that they account for age-related effects in their models. But these aren't my concerns. Rather:

(1) The texts (these edited newspaper articles) could be more predictable for younger than older adults. If so, effects with older adults could simply be because people are less likely to predict less than more predictable words.

(2) The GPT-2 generated surprisal scores may correspond more closely to younger than older adult responses -- that is, its next word predictions may be more younger- than older-adult-like.

In my view, the authors have two choices: they could remove the discussion of age-related effects, or they could try to address BOTH (1) and (2).

As an aside, consider what we would conclude if we drew similar conclusions from a study in which children and adults read the same (children's) texts, but we didn't test what was predictable to each of them separately.

The paper is really strong in other respects and if my concern is not addressed, the conclusions about age might be generally accepted.

---

## [Author Response]

The following is the authors’ response to the original reviews.

**Reviewer #1 (Public review):**
This manuscript reports a dual-task experiment intended to test whether language prediction relies on executive resources, using surprisal-based measures of predictability and an n-back task to manipulate cognitive load. While the study addresses a question under debate, the current design and modelling framework fall short of supporting the central claims. Key components of cognitive load, such as task switching, word prediction vs integration, are not adequately modeled. Moreover, the weak consistency in replication undermines the robustness of the reported findings. Below unpacks each point.Cognitive load is a broad term. In the present study, it can be at least decomposed into the following components:(1) Working memory (WM) load: news, color, and rank.(2) Task switching load: domain of attention (color vs semantics), sensorimotor rules (c/m vs space).(3) Word comprehension load (hypothesized against): prediction, integration.The components of task switching load should be directly included in the statistical models. Switching of sensorimotor rules may be captured by the "n-back reaction" (binary) predictor. However, the switching of attended domains and the interaction between domain switching and rule complexity (1-back or 2-back) were not included. The attention control experiment (1) avoided useful statistical variation from the Read Only task, and (2) did not address interactions. More fundamentally, task-switching components should be directly modeled in both performance and full RT models to minimize selection bias. This principle also applies to other confounding factors, such as education level. While missing these important predictors, the current models have an abundance of predictors that are not so well motivated (see later comments). In sum, with the current models, one cannot determine whether the reduced performance or prolonged RT was due to affecting word prediction load (if it exists) or merely affecting the task switching load.The entropy and surprisal need to be more clearly interpreted and modeled in the context of the word comprehension process. The entropy concerns the "prediction" part of the word comprehension (before seeing the next word), whereas surprisal concerns the "integration" part as a posterior. This interpretation is similar to the authors writing in the Introduction that "Graded language predictions necessitate the active generation of hypotheses on upcoming words as well as the integration of prediction errors to inform future predictions [1,5]." However, the Results of this study largely ignored entropy (treating it as a fixed effect) and only focus on surprisal without clear justification.In Table S3, with original and replicated model fitting results, the only consistent interaction is surprisal x age x cognitive load [2-back vs. Reading Only]. None of the two-way interactions can be replicated. This is puzzling and undermines the robustness of the main claims of this paper.
**Reviewer #2 (Public review):**
SummaryThis paper considers the effects of cognitive load (using an n-back task related to font color), predictability, and age on reading times in two experiments. There were main effects of all predictors, but more interesting effects of load and age on predictability. The effect of load is very interesting, but the manipulation of age is problematic, because we don't know what is predictable for different participants (in relation to their age). There are some theoretical concerns about prediction and predictability, and a need to address literature (reading time, visual world, ERP studies).Strengths/weaknessesIt is important to be clear that predictability is not the same as prediction. A predictable word is processed faster than an unpredictable word (something that has been known since the 1970/80s), e.g., Rayner, Schwanenfluegel, etc. But this could be due to ease of integration. I think this issue can probably be dealt with by careful writing (see point on line 18 below). To be clear, I do not believe that the effects reported here are due to integration alone (i.e., that nothing happens before the target word), but the evidence for this claim must come from actual demonstrations of prediction.The effect of load on the effects of predictability is very interesting (and also, I note that the fairly novel way of assessing load is itself valuable). Assuming that the experiments do measure prediction, it suggests that they are not cost-free, as is sometimes assumed. I think the researchers need to look closely at the visual world literature, most particularly the work of Huettig. (There is an isolated reference to Ito et al., but this is one of a large and highly relevant set of papers.)There is a major concern about the effects of age. See the Results (161-5): this depends on what is meant by word predictability. It's correct if it means the predictability in the corpus. But it may or may not be correct if it refers to how predictable a word is to an individual participant. The texts are unlikely to be equally predictable to different participants, and in particular to younger vs. older participants, because of their different experiences. To put it informally, the newspaper articles may be more geared to the expectations of younger people. But there is also another problem: the LLM may have learned on the basis of language that has largely been produced by young people, and so its predictions are based on what young people are likely to say. Both of these possibilities strike me as extremely likely. So it may be that older adults are affected more by words that they find surprising, but it is also possible that the texts are not what they expect, or the LLM predictions from the text are not the ones that they would make. In sum, I am not convinced that the authors can say anything about the effects of age unless they can determine what is predictable for different ages of participants. I suspect that this failure to control is an endemic problem in the literature on aging and language processing and needs to be systematically addressed.Overall, I think the paper makes enough of a contribution with respect to load to be useful to the literature. But for discussion of age, we would need something like evidence of how younger and older adults would complete these texts (on a word-by-word basis) and that they were equally predictable for different ages. I assume there are ways to get LLMs to emulate different participant groups, but I doubt that we could be confident about their accuracy without a lot of testing. But without something like this, I think making claims about age would be quite misleading.

We thank both reviewers for their constructive feedback and for highlighting areas where our theoretical framing and analyses could be clarified and strengthened. We have carefully considered each of the points raised and made substantial additions and revisions.

As a summary, we have directly addressed the concerns raised by the reviewers by incorporating task-switching predictors into the statistical models, paralleling our focus on surprisal with a full analysis and interpretation of entropy, clarifying the robustness (and limitations) of the replicated findings, and addressing potential limitations in our Discussion.

We believe these revisions substantially strengthen the manuscript and improve the reading flow, while also clarifying the scope of our conclusions. We will not illustrate these changes in more detail:

(1) Cognitive load and task-switching components.

We agree that cognitive load is a multifaceted construct, particularly since our secondary task broadly targets executive functioning. In response to Reviewer 1, we therefore examined task-switching demands more closely by adding the interaction term n-back reaction × cognitive load to a model restricted to 1-back and 2-back Dual Task blocks (as there were no n-back reactions in the Reading Only condition). This analysis showed significantly longer reading times in the 2-back than in the 1back condition, both for trials with and without an n-back reaction. Interestingly, the difference between reaction and no-reaction trials was smaller in the 2-back condition (β = -0.132, t(188066.09) = -34.269, p < 0.001), which may simply reflect the general increase in reading time for all trials so that the effect of the button press time decreases in comparison to the 1-back. In that sense, these findings are not unexpected and largely mirror the main effect of cognitive load. Crucially, however, the three-way interaction of cognitive load, age, and surprisal remained robust (β = 0.00004, t(188198.86) = 3.540, p < 0.001), indicating that our effects cannot be explained by differences in taskswitching costs across load conditions. To maintain a streamlined presentation, we opted not to include this supplementary analysis in the manuscript.

(2) Entropy analyses.

Reviewer 1 pointed out that our initial manuscript placed more emphasis on surprisal. In the revised manuscript, we now report a full set of entropy analyses in the supplementary material. In brief, these analyses show that participants generally benefit from lower entropy across cognitive load conditions, with one notable exception: young adults in the Reading Only condition, where higher entropy was associated with faster reading times. We have added these results to the manuscript to provide a more complete picture of the prediction versus integration distinction highlighted in the review (see sections “Control Analysis: Disentangling the Effect of Cognitive Load on Pre- and PostStimulus Predictive Processing” in the Methods and “Disentangling the Effect of Cognitive Load on Pre- and Post-Stimulus Predictive Processing“ in the Results).

(3) Replication consistency.

Reviewer 1 noted that the results of the replication analysis were somewhat puzzling. We take this point seriously and agree that the original model was likely underpowered to detect the effect of interest. To address this, we excluded the higher-level three-way interaction of age, cognitive load, and surprisal, focusing instead on the primary effect examined in this paper: the modulatory influence of cognitive load on surprisal. Using this approach, we observed highly consistent results between the original online subsample and the online replication sample.

(4) Potential age bias in GPT-2.

We thank Reviewer 2 for their thoughtful and constructive feedback and agree that a potential age bias in GPT-2’s next-token predictions warrants caution. We thus added a section in the Discussion explicitly considering this limitation, and explain why it should not affect the implications of our study.

**Reviewer #1 (Recommendations for the authors):**
The d-prime model operates at the block level. How many observation goes into the fitting (about 175*8=1050)? How can the degrees of freedom of a certain variable go up to 188435?

We thank the reviewer for spotting this issue. Indeed, there was an error in our initial calculations, which we have now corrected in the manuscript. Importantly, the correction does not meaningfully affect the results for the analysis of d-primes or the conclusions of the study (see line 102).

“A linear mixed-effects model revealed n-back performance declined with cognitive load (β = -1.636, t(173.13) = -26.120, p < 0.001), with more pronounced effects with advancing age (β = -0.014, t(169.77) = -3.931, p > 0.001; Fig. 3b, Table S1)”.

Consider spelling out all the "simple coding schemes" explicitly.

We thank the reviewer for this helpful suggestion. In the revised manuscript, we have now included the modelled contrasts in brackets after each predictor variable.

“Example from line 527: In both models, we included recording location (online vs. lab), cognitive load (1-back and 2back Dual Task vs. Reading Only as the reference level) and continuously measured age (centred) in both models as well as the interaction of age and cognitive load as fixed effects”.

The relationship between comprehension accuracy and strategies for color judgement is unclear or not intuitive.

We thank the reviewer for this helpful comment. The n-back task, which required participants to judge colours, was administered at the single-trial level, with colours pseudorandomised to prevent any specific colour - or sequence of colours - from occurring more frequently than others. In contrast, comprehension questions were presented at the end of each block, meaning that trial-level stimulus colour was unrelated to accuracy on the block-level comprehension questions. However, we agree that this distinction may not have been entirely clear, and we have now added a brief clarification in the Methods section to address this point (see line 534):

“Please note that we did not control for trial-level stimulus colour here. The n-back task, which required participants to judge colours, was administered at the single-trial level, with colours pseudorandomised to prevent any specific colour - or sequence of colours - from occurring more frequently than others. In contrast, comprehension questions were presented at the end of each block, meaning that trial-level stimulus colour was unrelated to accuracy on the blocklevel comprehension questions”.

Could you explain why comprehension accuracy is not modeled in the same way as d-prime, i.e., with a similar set of predictors?

This is a very good point. After each block, participants answered three comprehension questions that were intentionally designed to be easy: they could all be answered correctly after having read the corresponding text, but not by common knowledge alone. The purpose of these questions was primarily to ensure participants paid attention to the texts and to allow exclusion of participants who failed to understand the material even under minimal cognitive load. As comprehension accuracy was modelled at the block level with 3 questions per block, participants could achieve only discrete scores of 0%, 33.3%, 66.7%, or 100%. Most participants showed uniformly high accuracy across blocks, as expected if the comprehension task fulfilled its purpose. However, this limited variance in performance caused convergence issues when fitting a comprehension-accuracy model at the same level of complexity as the d′ model. To model comprehension accuracy nonetheless, we therefore opted for a reduced model complexity in this analysis.

RT of previous word: The motivations described in the Methods, such as post-error-slowing and sequential modulation effects, lack supporting evidence. The actual scope of what this variable may account for is unclear.

We are happy to elaborate further regarding the inclusion of this predictor. Reading times, like many sequential behavioural measures, exhibit strong autocorrelation (Schuckart et al., 2025, doi: 10.1101/2025.08.19.670092). That is, the reading time of a given word is partially predictable from the reading time of the previous word(s). Such spillover effects can confound attempts to isolate trialspecific cognitive processes. As our primary goal was to model single-word prediction, we explicitly accounted for this autocorrelation by including the log reading time of the preceding trial as a covariate. This approach removes variance attributable to prior behaviour, ensuring that the estimated effects reflect the influence of surprisal and cognitive load on the current word, rather than residual effects of preceding trials. We now added this explanation to the manuscript (see line 553):

“Additionally, it is important to consider that reading times, like many sequential behavioural measures, exhibit strong autocorrelation (Schuckart et al., 2025), meaning that the reading time of a given word is partially predictable from the reading time of the previous word. Such spillover effects can confound attempts to isolate trial-specific cognitive processes. As our primary goal was to model single-word prediction, we explicitly accounted for this autocorrelation by including the reading time of the preceding trial as a covariate”.

Block-level d-prime: It was shown with the d-prime performance model that block-level d-prime is a function of many of the reading-related variables. Therefore, it is not justified to use them here as "a proxy of each participant's working memory capacity."

We thank the reviewer for their comment. We would like to clarify that the d-prime performance model indeed included only dual-task d-primes (i.e., d-primes obtained while participants were simultaneously performing the reading task). In contrast, the predictor in question is based on singletask d-primes, which are derived from the n-back task performed in isolation. While dual- and singletask d-primes may be correlated, they capture different sources of variance, justifying the use of single-task d-primes here as a measure of each participant’s working memory capacity.

Word frequency is entangled with entropy and surprisal. Suggest removal.

We appreciate the reviewer’s comment. While word frequency is correlated with word surprisal, its inclusion does not affect the interpretation of the other predictors and does not introduce any bias. Moreover, it is a theoretically important control variable in reading research. Since we are interested in the effects of surprisal and entropy beyond potential biases through word length and frequency, we believe these are important control variables in our model. Moreover, checks for collinearity confirmed that word frequency was neither strongly correlated with surprisal nor entropy. In this sense, including it is largely pro forma: it neither harms the model nor materially changes the results, but it ensures that the analysis appropriately accounts for a well-established influence on word processing.

Entropy reflects the cognitive load of word prediction. It should be investigated in parallel and with similar depth as surprisal (which reflects the load of integration).

This is an excellent point that warrants further investigation, especially since the previous literature on the effects of entropy on reading time is scarce and somewhat contradictory. We have thus added additional analyses and now report the effects of cognitive load, entropy, and age on reading time (see sections “Disentangling the Effect of Cognitive Load on Pre- and Post-Stimulus Predictive Processing” in the Results, “Control Analysis: Disentangling the Effect of Cognitive Load on Pre- and Post-Stimulus Predictive Processing” in the Methods as well as Fig. S7 and Table S6 in the Supplements for full results). In brief, we observe a significant three-way interaction among age, cognitive load, and entropy. Specifically, while all participants benefit from low entropy under high cognitive load, reflected by shorter reading times, in the baseline condition this benefit is observed only in older adults. Interestingly, in the baseline condition with minimal cognitive load, younger adults even show a benefit from high entropy. Thus, although the overall pattern for entropy partly mirrors that for surprisal – older adults showing increased reading times when word entropy is high and generally greater sensitivity to entropy variations – the effects differ in one important respect. Unlike for surprisal, the detrimental impact of increased word entropy is more pronounced under high cognitive load across all participants.

**Reviewer #2 (Recommendations for the authors):**
I agree in relation to prediction/load, but I am concerned (actually very concerned) that prediction needs to be assessed with respect to age. I suspect this is one reason why there is so much inconsistency in the effects of age in prediction and, indeed, comprehension more generally. I think the authors should either deal with it appropriately or drop it from the manuscript.

Thank you for raising this important concern. It is true that prediction is a highly individual, complex process as it depends upon the experiences a person has made with language over their lifespan. As such, one-size-fits-all approaches are not sufficient to model predictive processing. In our study, we thus took particular care to ensure that our analyses captured both age-related and other interindividual variability in predictive processing.

First, in our statistical models, we included age not only as a nuisance regressor, but also assessed age-related effects in the interplay of surprisal and cognitive load. By doing so, we explicitly model potential age-related differences in how individuals of different ages predict language under different levels of cognitive load.

Second, we hypothesised that predictive processing might also be influenced by a range of interindividual factors beyond age, including language exposure, cognitive ability, and more transient states such as fatigue. To capture such variability, all models included by-subject random intercepts and slopes, ensuring that unmodelled individual differences were statistically accommodated.

Together, these steps allow us to account for both systematic age-related differences and residual individual variability in predictive processing. We are therefore confident that our findings are not confounded by unmodelled age-related variability.

Line 18, do not confuse prediction (or pre-activation) with predictability. Predictability effects can be due to integration difficulty. See Pickering and Gambi 2018 for discussion. The discussion then focuses on graded parallel predictions, but there is also a literature concerned with the prediction of one word, typically using the "visual world" paradigm (which is barely cited - Reference 60 is an exception). In the next paragraph, I would recommend discussing the N400 literature (particularly Federmeier). There are a number of reading time studies that investigate whether there is a cost to a disconfirmed prediction - often finding no cost (e.g., Frisson, 2017, JML), though there is some controversy and apparent differences between ERP and eye-tracking studies (e.g., Staub). This literature should be addressed. In general, I appreciate the value of a short introduction, but it does seem too focused on neuroscience rather than the very long tradition of behavioural work on prediction and predictability.

We thank the reviewer for this suggestion. In the revised manuscript, we have clarified the relevant section of the introduction to avoid confusion between predictability and predictive processing, thereby improving conceptual clarity (see line 16).

“Instead, linguistic features are thought to be pre-activated broadly rather than following an all-or-nothing principle, as there is evidence for predictive processing even for moderately- or low-restraint contexts (Boston et al., 2008; Roland et al., 2012; Schmitt et al., 2021; Smith & Levy, 2013)”.

We also appreciate the reviewer’s comment regarding the introduction. While our study is behavioural, we frame it in a neuroscience context because our findings have direct implications for understanding neural mechanisms of predictive processing and cognitive load. We believe that this framing is important for situating our results within the broader literature and highlighting their relevance for future neuroscience research.

I don't think 2 two-word context is enough to get good indicators of predictability. Obviously, almost anything can follow "in the", but the larger context about parrots presumably gives a lot more information. This seems to me to be a serious concern - or am I misinterpreting what was done?

This is a very important point and we thank the reviewer for raising it. Our goal was to generate word surprisal scores that closely approximate human language predictions. In the manuscript, we report analyses using a 2-word context window, following recommendations by Kuribayashi et al. (2022).

To evaluate the impact of context length, we also tested longer windows of up to 60 words (not reported). While previous work (Goldstein et al., 2022) shows that GPT-2 predictions can become more human-like with longer context windows, we found that in our stimuli – short newspaper articles of only 300 words – surprisal scores from longer contexts were highly correlated with the 2word context, and the overall pattern of results remained unchanged. To illustrate, surprisal scores generated with a 10-word context window and surprisal scores generated with the 2-word context window we used in our analyses correlated with Spearman’s ρ = 0.976.

Additionally, on a more technical note, using longer context windows reduces the number of analysable trials, since surprisal cannot be computed for the first k words of a text with a k-word context window (e.g., a 50-word context would exclude ~17% of the data).

Importantly, while a short 2-word context window may introduce additional noise in the surprisal estimates, this would only bias effects toward zero, making our analyses conservative rather than inflating them. Critically, the observed effects remain robust despite this conservative estimate, supporting the validity of our findings.

However, we agree that this is a particularly important and sensitive point, and have now added a discussion of it to the manuscript (see line 476).

“Entropy and surprisal scores were estimated using a two-word context window. While short contexts have been shown to enhance GPT-2’s psychometric alignment with human predictions, making next-word predictions more human-like (Kuribayashi et al., 2022), other work suggests that longer contexts can also increase model–human similarity (Goldstein et al., 2022). To reconcile these findings in our stimuli and guide the choice of context length, we tested longer windows and found surprisal scores were highly correlated with the 2-word context (e.g., 10-word vs. 2-word context: Spearman’s ρ = 0.976), with the overall pattern of results unchanged. Additionally, employing longer context windows would have also reduced the number of analysable trials, since surprisal cannot be computed for the first k words of a text with a k-word context window. Crucially, any additional noise introduced by the short context biases effect estimates toward zero, making our analyses conservative rather than inflating them”.

Line 92, task performance, are there interactions? Interactions would fit with the experimental hypotheses.

Yes, we did include an interaction term of age and cognitive load and found significant effects on nback task performance (d-primes; b = -0.014, t(169.8) = -3.913, p < 0.001), but not on comprehension question accuracy (see table S1 and Fig. S2 in the supplementary material).

Line 149, what were these values?

We found surprisal values ranged between 3.56 and 72.19. We added this information in the manuscript (see line 143).